

Observed aerosol suppression of cloud ice in low-level Arctic mixed-phase clouds

Matthew S. Norgren[1], Gijs de Boer[1,2], Matthew D. Shupe[1,2]

[1]Cooperative Institute for Research in Environmental Sciences, University of Colorado, Boulder, CO, USA
[2]Physical Sciences Division, Earth System Research Laboratory, National Oceanic and Atmospheric Administration,
Boulder, CO, USA

*Correspondence to*: Matthew S. Norgren (matthew.norgren@colorado.edu)

**Abstract.** The interactions that occur between aerosols and a mixed-phase cloud system, and the subsequent alteration of the
microphysical state of such clouds, is a problem that has yet to be well constrained. Advancing our understanding of aerosol-
ice processes is necessary to determine the impact of natural and anthropogenic emissions on Earth's climate and to improve
our capability to predict future climate states. This paper deals specifically with how aerosols influence ice mass production
in low-level Arctic mixed-phase clouds. In this study, a 9-year record of aerosol, cloud and atmospheric state properties is
used to quantify aerosol influence on ice production in mixed-phase clouds. It is found that mixed-phase clouds present in a
clean aerosol state have higher ice water content by a factor of 1.22 to 1.63 at cloud base than do similar clouds in cases with
higher aerosol loading. We additionally analyze radar-derived mean Doppler velocities to better understand the drivers
behind this relationship, and conclude that aerosol suppression of ice nucleation, together with reduced riming rates in
polluted clouds are likely influences on the observed reductions in IWC.

## 1 Introduction

Surface temperatures in the Arctic are rising in response to increases in radiative forcings. The rate of warming in the Arctic
is significantly higher than the mean rate of temperature increase for the globe (Manabe and Stouffer, 1980; Navarro et al.,
2016). This warming has consequences to the physical and ecological systems of the Arctic environment, and these impacts
are expected to become more severe in the future (Stroeve et al., 2008; Swart, 2017; Jay et al., 2011; Hinzman et al., 2013;
Bindoff et al., 2013). These changes to the Arctic system have implications for biological and human activity in the region.

Numerous feedback mechanisms have been proposed as drivers of the observed amplified surface warming signal
in the Arctic (Serreze et al., 2009). Modeling studies have indicated that surface-albedo and temperature feedbacks are the
main mechanisms responsible (Serreze and Francis, 2006; Screen and Simmonds, 2010; Taylor et al., 2013; Pithan and
Mauritsen, 2014). Yet limitations of models, including the required treatment of clouds through sub-grid parameterizations,
leave gaps in our understanding of the role that clouds play in regulating the Arctic surface temperature response. Clouds are
a prevalent and critical contributor to these central feedback processes because of the role they play in modulating the flux of
energy to the surface. The micro- and macrophysical properties of clouds influence the thermodynamic and radiative
properties of the atmosphere (Curry and Ebert, 1992; Pinto 1998; Shupe et al., 2011). The net impact of a cloud on the





surface energy budget is strongly dependent on, among other factors, the phase of the water of which it is composed (Shupe and Intrieri, 2004). Cloud phase also impacts precipitation characteristics and is a factor in cloud lifetime, which is another relevant parameter governing how clouds fit into the Arctic climate system. Understanding phase partitioning in clouds is therefore critical, but an incomplete view of key microphysical processes, including ice nucleation, inhibits such

understanding (Prenni et al., 2007).

The thermodynamic conditions (e.g., temperature and supersaturation) available for cloud formation in the troposphere necessitate aerosols be present for cloud development to occur. Thus, aerosols are a fundamental component of mixed-phase clouds, acting as cloud condensation nuclei (CCN) and ice nucleating particles (INP). In the Arctic, aerosol concentrations follow a seasonal cycle with a high number of aerosol particles transported to the region from mid-latitudes in

winter and spring. This phenomenon, known as Arctic haze, results from accumulation of transported particles in a thermodynamically stable environment, where precipitation and chemical reactions are both limited due to the cold Arctic night (Barrie, 1986; Shaw, 1995; Quinn et al., 2007; Law et al., 2014). Yet understanding the relevance these aerosols have to Arctic cloud processes is difficult because of our limited understanding of the aerosol composition, size, and vertical distribution present. For example, scarcity of INP (Bigg, 1996) is a significant limitation on ice mass production and is a

feature of the Arctic environment used to explain the long persistence times of mixed-phase clouds (Pinto, 1998; Harrington et al., 1999). Additionally, INP concentrations have been shown to vary greatly in time and space in the Arctic environment (Fountain and Ohtake, 1985; Rogers et al., 2001), and the development of INP parameterizations based on limited observational data has proven to be challenging (DeMott et al., 2010; DeMott et al., 2015). This inadequate understanding of INP properties has led to difficulties in modelling ice-containing clouds. Arctic aerosol composition is an equally murky

problem. Quinn et al. (2002), have shown that aerosol composition varies significantly throughout the year, with sulfate-coated particles being highly prevalent in spring. Still, a proper representation of aerosol concentrations and information on composition in and around mixed-phase cloud systems is lacking.

That being said, several aerosol-cloud effects have been detected in mixed-phase cloud systems: the first and second aerosol indirect effects have been observed (Lohman and Feichter, 2005). Several observational studies have found evidence

for aerosol impacts on Arctic mixed-phase clouds. Using surface-based sensors at Barrow, both Garrett and Zhao (2006) and Lubin and Vogelmann (2006) showed that a reduction of droplet size associated with elevated aerosol particle concentrations results in elevated emissivity of the cloud layer, thereby significantly increasing longwave radiation at the surface and contributing to warming. Lance et al. (2011) used in situ data from Arctic clouds to show that CCN concentrations, through the first indirect effect and riming indirect effect, may have a stronger influence on ice production than do INP

concentrations. These past studies suggest that further interrogation of aerosol alterations to the microphysical state of mixed-phase clouds systems is warranted.

Ultimately, our incapacity to understand the physics driving cloud systems hampers our ability to evaluate future climate states. Global circulation models (GCMs) allow us to assess the Earth system response to a variety of climate forcing scenarios. Even though such models are routinely invoked for guiding policy and scientific understanding, they are





constrained by their inability to represent certain physical processes. Limited computational power requires sub-grid parameterizations of clouds and cloud processes that often do not represent reality. The representation of clouds and cloud phase requires substantial improvement, with both temperature-dependent and prognostic phase partitioning schemes having been demonstrated to be inadequate. For example, Cesana et al. (2015) determined that even with state-of-the-art prognostic

cloud microphysics, models such as CAM5 and HadGEM still had significant biases in the representation of ice clouds. Such biases result in models having significant surface temperature errors, such as those found over Greenland's ice sheet in CAM5 (Kay et al., 2016). The impacts of these model limitations become particularly clear in sensitive parts of the world, such as the Arctic, where there is significant variability in cloud phase. Improved understanding of cloud processes can help to alleviate GCM shortcomings.

In this paper, we aim to demonstrate that aerosol alterations of cloud liquid properties are a significant control on ice production in Arctic mixed-phase clouds. This includes aerosol influences on both nucleation of ice crystals and secondary ice mass growth processes. To do this, we utilize a 9-year record of radar, microwave radiometer, and radiosonde measurements from the US Department of Energy (DOE) Atmospheric Radiation Measurement (ARM) Program facility in Utqiaġvik (formerly Barrow), Alaska, along with aerosol measurements made by the National Oceanic and Atmospheric

Administration (NOAA) Global Monitoring Division (GMD) to evaluate relationships between cloud ice water content (IWC) and aerosol concentrations near the surface. In the following sections, we first provide an overview of the instruments and methods used in this study. This is followed by observational results and a discussion of these results and their impact on our understanding of cloud ice production.

**2 Data and methods**

A multi-sensor method is used to identify stratiform mixed-phase clouds that are the subject of this study. These clouds are characterized by having shallow liquid layers, the tops of which are at heights less than 2 km from the surface. The clouds may or may not be precipitating to the surface. Radar and other remote sensing tools are used to characterize ice and liquid properties of these cloud layers. Ground-based measurements of aerosol scattering coefficients are used to approximate the

aerosol loading of the lower atmosphere. Finally, radiosondes, in combination with ground based remote sensors and model output, are used to classify the thermodynamic state of the atmosphere during cloudy periods.

     Sampling took place at the ARM North Slope of Alaska (NSA) site, located just to the northeast of Utqiaġvik, Alaska (71.323N, 156.616W). This site is ideal for this study because it features a high occurrence of mixed-phase clouds (Shupe et al., 2011) and provides an extensive data set from which to develop adequate statistics for deriving relationships of

interest. Here, we use the 9-year period from January 2000 to December 2008.





**2.1 Cloud Properties**

Vertical profiles of radar reflectivity and retrieved IWC are based on reflected power measured by a vertically-pointing Ka-band 35 GHz millimeter cloud radar (MMCR; Moran et al., 1998; Kollias et al., 2007). The MMCR product used here provides at 45-meter vertical resolution and 10-second temporal resolution. Five minute averages of the reflectivity are used

to estimate IWC using an empirically derived power-law relationship:

$$\text{IWC} = a Z^b \tag{1}$$

Here, $Z$ is the measured returned power to the radar. The coefficients, $a$ and b are seasonally adjusted tuning parameters

based on observations made during the Surface Heat Budget of the Arctic Ocean (SHEBA) experiment (Shupe et al., 2005), which took place over a full annual cycle to the north of Utqiaġvik in the Beaufort and Chukchi Seas. Uncertainties of up to 100 percent in the retrieved IWC values arise from variability in the ice crystal size distribution and crystal habit that are not captured by the instantaneous value of the $a$ parameter in the power-law (Shupe et al., 2005). The seasonal variability of the coefficients partially accounts for the temperature and aerosol related crystal habit dependences of the IWC retrieval, though

this empirical method lacks the resolution needed to capture variability on sub-monthly time scales. Additionally, while occurring in the same quadrant of the Arctic, the SHEBA experiment occurred in a meteorological environment that may be different from the one in which the NSA site is situated. For one, SHEBA took place far from land masses, whereas NSA is situated at the coastal boundary. It is expected that this difference would result in variability in aerosol, thermodynamic and radiative atmospheric states, all of which could impact ice particle properties. Given these limitations, how well this

SHEBA-based retrieval can be used to represent ice properties at the NSA site is difficult to quantify. Therefore, we also present the corresponding radar reflectivity data, which is not bound by the same limitations as the IWC retrievals, with a change in reflectivity being qualitatively indicative of a shift in the cloud ice properties.

We use ice crystal fall speed ($V_f$) in conjunction with the IWC retrievals to make inferences about ice crystal number and mean size. The second moment of the MMCR is the mean Doppler velocity (MDV), which characterizes the

motion of atmospheric hydrometeors. The vapor deposition process typically promotes ice crystal growth to sizes larger than liquid drops, and hence the radar reflectivity signal is generally dominated by ice crystals in the sampled volume of a mixed-phase cloud. The MDV is therefore representative of ice crystal motions, which are governed by gravity, small-scale air motions within the cloud layer (i.e., eddy motions that result from convective processes within the cloud), and synoptic scale motions of the cloud. Here we assume that variability of synoptic scale motions occurs over timescales much longer than the

in-cloud eddy motions, and that synoptic motions are at least an order of magnitude less than ice crystal fall speeds. Time averaging of MDV on timescales longer than the cloud eddy timescale allows us to remove the eddy influence on ice crystal motion (Orr and Kropfli, 1999). Doing so yields average ice crystal motions resulting from gravitational force -- the mean fall speed, $V_f$. In this study, we used a 120-minute time averaging window to calculate $V_f$. The appropriate time averaging




window is a subjective decision, although $V_f$ does not depend strongly on the averaging time as long as a stable portion of the cloud layer (i.e. averaged points do not included values from out of cloud) is sampled (Orr and Kropfli, 1999).

We use liquid water path (LWP) to classify the amount of liquid water in the mixed-phase cloud. This classification is done to control for environmental influence on cloud liquid water, which can interact to form ice within the cloud. That is,

we are interested in aerosol effects on clouds for different cloud system types as defined by the LWP of the cloud. By comparing cloud ice properties for narrow LWP values, distinguishing between the microphysical differences that exist among clean and polluted clouds becomes possible. LWP is derived from brightness temperature measurements at 23.8 and 31.4 GHz from a microwave radiometer (Turner et al., 2007). When the physical method for retrieving LWP was not available, a variable coefficient, bilinear, statistical method is used (Liljegren et al., 2001). Respectively, these are the ARM

MWRRET and MWRLOS retrievals.

Cloud top height is inferred from radar reflectivity profiles and is defined to be the height of the highest radar return of the low-level cloud, similar to the method of Moran et al. (1998). Cloud base height is defined at the bottom of the liquid-containing layer, which is determined from 905 nm Vaisala ceilometer measurements (15 m vertical resolution). In clouds devoid of liquid water or with intense precipitation, the ceilometer backscatter signal does not clearly define a cloud base

height. In these cases, the discontinuity point in this ceilometer signal is used to identify cloud base (Shupe et al., 2013).

### 2.2 Aerosol Measurements

We use one minute averaged values of scattering coefficient at 550 nm, which are measured at the surface by a TSI nephelometer deployed as part of the aerosol observing system (AOS). These surface-based measurements are used to approximate aerosol concentrations in the cloud layer. Scattering coefficients have been used to identify atmospheric aerosol

loading in past studies because cloud-relevant aerosols are often efficient at scattering 550 nm light (Garrett et al., 2004; Garrett and Zhao, 2006). The AOS did not continuously operate over the 9-year period, causing numerous periods with missing scattering coefficient data. Linear interpolation is used between scattering coefficient measurements separated by less than 24 hours to infer the scattering coefficient value at one-minute intervals to match the time and resolution of the IWC profiles derived from the MMCR. If the sampling time for any given IWC profile is more than 24 hours from the

nearest aerosol data point, the profile is not used in this study.

We separate the dataset into clean and polluted regimes to study the aerosol impact on cloud IWC. For the remainder of this paper, polluted conditions are defined to be the top 30 percent of recorded scattering coefficient values, and clean condition the lowest 30 percent of scattering coefficients for the set of cloud IWC profiles under study. The middle 40% of the data are not considered.



### 2.3 Environmental conditions

Radiosondes were launched by both the DOE ARM program and the National Weather Service (NWS) office in Utqiaġvik at a frequency of one to four times per day over the course of the MMCR data record. These radiosonde measurements are used to evaluate temperature and supersaturation with respect to ice and liquid within the cloud layer. Because the radiosondes

were launched at 6- or 12-hourly increments, we use the DOE ARM MERGESONDE value-added product to obtain information for the time periods between the balloon flights. This product combines radiosonde, ground-based remote sensor and forecasting model data to interpolate temperature and humidity fields between radiosonde profiles (Troyan, 2012). We expect dry biasing errors to be minimal in both the radiosonde and mergesonde data sets because the data used in this study comes from relatively warm and humid regions of the atmosphere (Fleming, 1998), and therefore we did not correct for

these effects. The corresponding temperature and humidity profiles for each IWC profile are identified from the mergesonde data. The maximum in-cloud relative humidity with respect to ice (RHi) and minimum temperature ($T_{min}$) within the cloud layer are used to classify clouds as mixed-phase and to ensure the presence of ice in the cloud. The IWC retrieval does not explicitly select for the presence of ice and there is a risk of contamination from liquid water at warmer temperatures. Limiting the study to clouds with $T_{min} < -6°C$ reduces contamination of the IWC retrieval by liquid water, though it is

possible that some of these clouds may still be lacking ice.

### 2.4 Vertical normalization of cloud variable profiles

To observe the effects of aerosol on IWC in mixed-phase clouds, we examine the shape of a mean IWC profile under polluted and clean aerosol conditions. To do so, we create vertical IWC profiles that are normalized in depth. Cloud base, the bottom of the liquid layer, is assigned a value of 0 and cloud top a value of 1. For each IWC profile the IWC values are

placed on a linear grid between 0 and 1, proportional to their fractional height above cloud base. The resolution of the normalized cloud grid is set so that it matches the number of sampled points by the radar of a 1km thick cloud, $\frac{1000m}{45m} = 23 bins$. Clouds thinner than 1km have less than 23 IWC values in their sampled profile and in these cases IWC values are linearly interpolated between grid points on the normalized grid. Normalized profiles of IWC are subsequently aggregated based on defined environmental criteria (see section 2.5). For aggregated subsets of classified IWC profiles, mean IWC

values are found for each normalized height to create a mean IWC profile. In addition to the IWC profiles, this aggregation and mean profile creation method is also applied to $V_f$ and reflectivity data to generate mean cloud profiles of these two variables.

### 2.5 Cloud classification and grouping

Radar reflectivity, IWC and $V_f$ parameters from a cloud layer are dependent on the combined state of numerous

environmental and cloud microphysical variables. In an attempt to account for environmental influences on the retrieved cloud properties, we group retrieved profiles by a defined set of corresponding environmental and physical properties.





We restrict this study to clouds with liquid layers less than 1 km thick. Cloud depth is an important parameter because it helps to define the scale of the interaction zone for liquid and ice particles, with deeper clouds having more opportunity to convert liquid water to ice. Additionally, deeper clouds tend to have stronger and more complex dynamics than do shallower clouds, which can obscure the view of aerosol influences on cloud ice. Cloud base height is arbitrarily

limited to below 2 km to increase the likelihood of coupling between the cloud and the surface, where the aerosol measurements occur. In this study, we do not explicitly require, or attempt to identify, coupling between the surface and cloud layer.

Clouds are required to have a maximum relative humidity with respect to ice ($RHi_{max}$) greater than 100 percent within the mixed-phase cloud layer. That is, some portion of the mixed-phase cloud layer must be saturated with respect to

10 ice. This requirement on ice saturation is a necessary (though not a sufficient) condition for ice nucleation within the cloud layer. Finally, seasonal differences in synoptic scale meteorological conditions and aerosol composition and type are controlled for by limiting the analysis to the months of December, January, February, March, April and May.

The liquid water layer depth and liquid water density are moderators of deposition and riming rates. To constrain the influence of liquid water on ice formation, we designate LWP regimes within which to compare cloud ice properties.

Clouds are sorted into five LWP bins -- $\{LWP0 = 0.00 - 10.00 \text{ g m}^{-2}; LWP1 = 10.00 - 23.82 \text{ g m}^{-2}; LWP2 = 23.82 - 63.25 \text{ g m}^{-2}; LWP3 = 63.25 - 126.84 \text{ g m}^{-2}; LWP4 = 126.84 - 994.50 \text{ g m}^{-2}\}$. The first bin is representative of ice clouds and clouds with very limited liquid water, as there is uncertainty on the order of 15 g m$^{-2}$ in the LWP retrieval resulting from instrument noise. The next four bin widths are spaced according to the 25th, 50th, 75th and 100th percentiles of the observed LWP distribution.

## 3 Results

### 3.1 Radar reflectivity

Figure 1 shows reflectivity in relation to normalized height in the cloud layer.



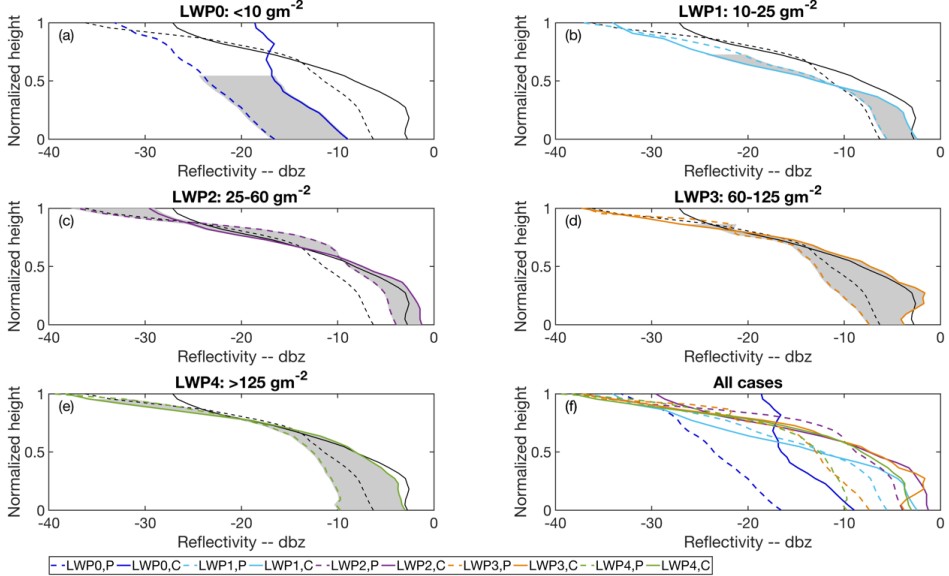

**Figure 1: Reflectivity profiles for the five LWP regimes for a) LWP0, b) LWP1 c) LWP2 d) LWP3 e) LWP4 and f) all LWP bins. 0 corresponds to cloud base and 1 corresponds to cloud top. Dashed lines represent polluted clouds, and solid lines indicate clean cases. The black lines, shown for reference in each panel, represent the mean profiles for the combined LWP bins. Grey shading**
**represents regions of the cloud layer where there is a statistically significant difference between the clean and polluted reflectivity distributions.**

Dashed lines represent mean reflectivity profiles from the aggregation of polluted cases and solid lines correspond to clean cases. The line color designates the LWP regime of the cloud. To determine if the difference between the clean and
10 polluted profiles is statistically significant, we perform an unequal variance t-test for each vertical bin of the normalized cloud layer. Interpolated reflectivity values arising from the height normalization process are included in samples input to the t-test. Throughout this paper, statistical significance is defined at a 95% confidence level. Statistically significant differences between clean and polluted profiles are indicated with grey shading.

At cloud top, reflectivity values are typically small due to the presence of small ice crystals. Reflectivity increases
with decreasing height in the cloud layer as ice mass growth occurs due to deposition and riming. These increases in reflectivity are more prominent for clean cases than for polluted clouds in all LWP bins, such that reflectivity is larger near cloud base for clean clouds. This is indicative of a greater rate of ice mass growth through the column for clean clouds. There is not a linear response in reflectivity to LWP: The ice-dominated clouds (LWP0) most often have the lowest reflectivity values, relative to all other LWP bins, in the bottom half of the cloud layer. The highest reflectivity values near



cloud base are found in intermediate LWP cases (LWP1 and LWP2), while the highest LWP clouds have lower reflectivity values.

Radar reflectivity is a direct measurement made by the cloud-radar, and includes no assumptions about cloud microphysics, though it is dependent on the cloud properties. The segregation of reflectivity profiles presented in Figure 1 is

5   evidence for aerosol interactions within mixed-phase cloud systems. In the following subsections, we further examine cloud IWC and ice crystal fall speed profiles for insight into the details of these aerosol-ice interactions.

### 3.2 Cloud IWC

The observed reflectivity values are transformed to IWCs through the power-law method outlined in the Section 2. The mean IWC profiles are presented in Figure 2.

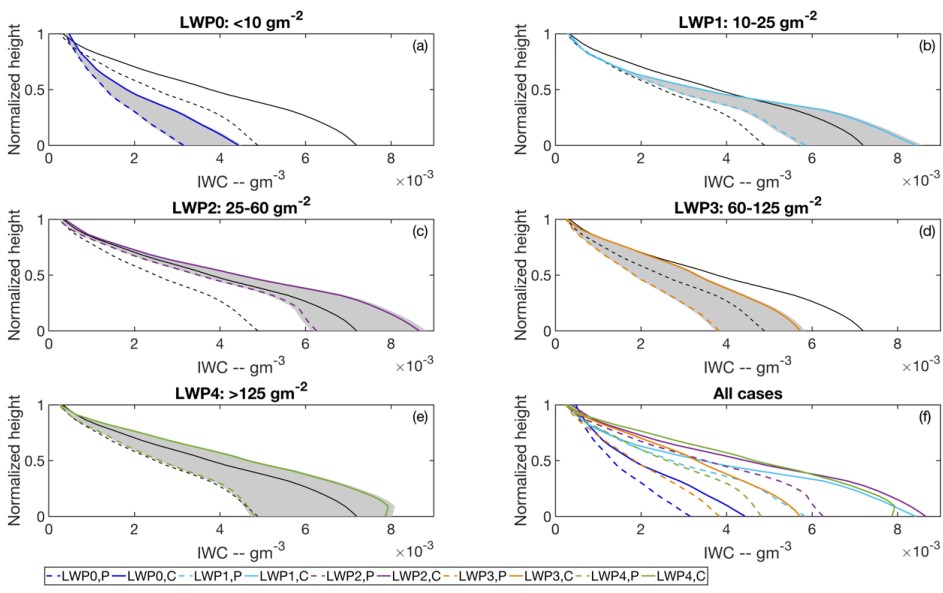

**Figure 2: Vertical profiles of ice water content for clean and polluted conditions, as in Fig. 1, for a) LWP0, b) LWP1 c) LWP2 d) LWP3 e) LWP4 and f) all LWP bins. The black reference lines represent the mean profiles for all the LWP bins. Grey shading represents regions of the cloud layer where there is a statistically significant difference between the clean and polluted IWC distributions.**

In all LWP bins, IWC values are less than 0.001 $g/m^3$ at cloud top. At cloud base, clean cloud IWCs tend to be a factor of 1.2-1.5 times greater than the IWCs in the corresponding polluted cases. All IWC profiles follow the same general shape, with low IWC values at cloud top followed by a fairly linear increase in IWC, which starts to decrease in roughly the bottom 10 percent of the cloud layer. This decrease in the rate of IWC increase near cloud base is likely due to the impacts of



less saturated air entraining into the bottom of the cloud, slowing growth processes in this region. For a given LWP regime, the clean clouds have a greater integrated column IWC, or ice water path.

The ordering of IWC at cloud base as a function of LWP is consistent with that of the cloud-base reflectivity. The exception is for polluted clouds, where LWP4 has a higher cloud base IWC than does LWP3, which is reverse of what is 5 found for reflectivity. While the ordering of the profiles is, more or less, consistent, the shape and relative positions of the lines varies between reflectivity and IWC. These inconsistencies between the two variables are caused by the seasonal nature of the IWC power-law retrieval. In later months (late spring) LWP tends to increase, while the $a$ coefficient of (1) decreases, and therefore the IWC for a given reflectivity decreases in these later months. This seasonal variation in the IWC retrieval can explain why LWP4 has a greater cloud base IWC than does LWP3: the IWC profile sample day of year distribution (Fig. 10 7) for LWP3 polluted clouds is more skewed towards later spring days than for LWP4.

Similar to the reflectivity profiles, statistical significance is determined by a 2-sample t-test at each vertical cloud layer. To account for uncertainty in the IWC retrieval, each profile is multiplied by an error factor, ranging between 0.5 to 2 (to account for an error of up to 100% of the IWC value), with the value probabilistically assigned based on a truncated Gaussian distribution over this range with a mean centered at 1. The same error factor is used for all values in a profile 15 because we expect ice crystal habit and size distribution variability to be the leading source of uncertainty in IWC retrieval (Shupe et al., 2005; Hong, 2007). Therefore, errors in retrieved IWC would be highly correlated within a profile. Populations of clean and polluted IWC profiles with the applied error factors are used in a t-test to produce a height profile of p-value. We then repeat this process 1000 times, with each test generating a new unique profile of p-values. To test for statistical significance, the set of 1000 p-value profiles is averaged, and the resulting mean p-value profile has each value compared 20 against a 95% significance level. The statically significant regions of the IWC profiles determined in this way are shaded in grey in Figure 2. The results show statistically significant differences at almost all heights for all LWP regimes.

### 3.3 Ice crystal fall speed, $V_f$

The vertical structure of mean ice crystal fall speed in the cloud layer indicates changes in the size, surface area to volume 25 ratio, crystal habit and crystal orientation. Generally, nucleation, deposition, aggregation and riming are the significant processes that change the ice mass to cross-sectional area relationship for a given cloud volume, and therefore variations in ice crystal fall speed are inherently linked to these processes. Ice crystal fall speed, in combination with IWC, allows us to infer relative information about ice crystal size and number properties at a given location in the cloud layer if we assume similar crystal habits and crystal orientations at each layer.





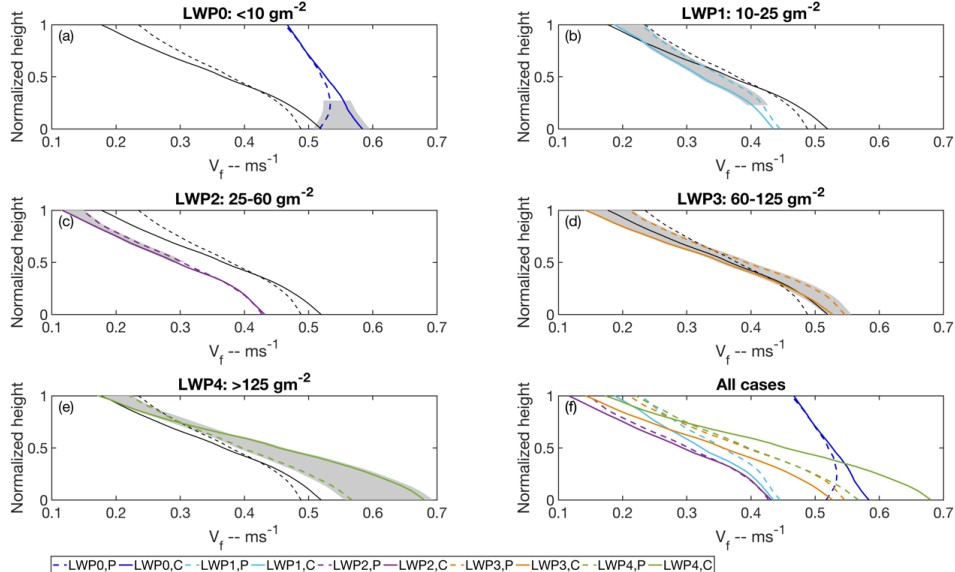

**Figure 3: Vertical profiles of 120-minute time averaged mean Doppler velocity for clean and polluted conditions for a) LWP0, b) LWP1 c) LWP2 d) LWP3 e) LWP4 and f) all LWP bins. The black reference lines in each panel represent the mean profiles for all the LWP bins. Grey shading represents regions of the cloud layer where there is a statistically significant difference between the clean and polluted $V_f$ distributions.**

Figure 3 shows the vertical fall speed profiles of all cloud cases. At cloud top, the mean fall speed of ice crystals for polluted clouds is greater than the mean fall speed for the clean cases in all LWP scenarios, except for ice clouds (LWP0 cases). Considering the equivalent IWCs at cloud top, the greater fall speeds in the polluted clouds indicates the presence of a larger mean crystal size, which must be matched with a reduction in ice crystal number, to drive the observed reflectivity/IWC response (a more detailed discussion is offered in Section 4.1/4.2). Alternatively, the fall speed variation could be the result of aerosol-induced changes in crystal habit and orientation, though we do not have evidence that these properties are influenced by INP concentrations.

The relationship between cloud layer depth and $V_f$ varies for different LWP bins. The LWP0 clouds have the least variation in $V_f$ with depth. LWP1,2,3,4 cases all have clean clouds with cloud top $V_f$ that is less than that of the corresponding polluted clouds. Moving lower in the cloud layer differences between clean and polluted $V_f$ is reduced – there is convergence of the fall speeds. The LWP4 cases are similar to that of LWP1,2,3 with the notable feature being high cloud base $V_f$ in clean clouds. The LWP4 bin is the only LWP regime, other than LWP0, where clean clouds have a greater $V_f$ than polluted clouds at cloud base that is statistically significant.



It is important to note that at cloud base, LWP1 and LWP2 clouds have the highest IWC values, yet these clouds have the lowest $V_f$ values. This implies that these cases contain clouds with a large amount of smaller ice crystals, or that these cases mainly consist of ice crystals with shapes that are large but slow falling.

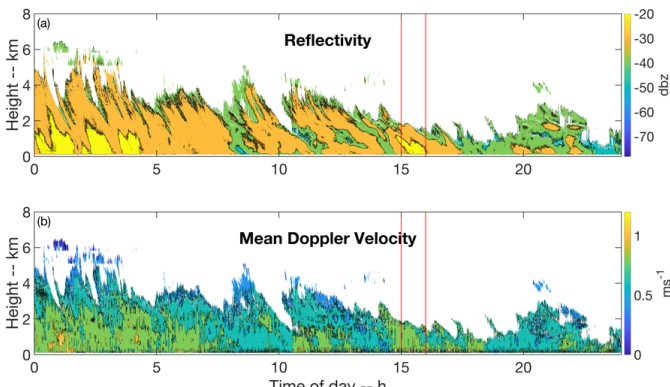

**Figure 4: (a) Reflectivity and (b) Mean Doppler Velocity from the MMCR for a LWP0 cloud on January 19th, 2006. Time from hours 15 to 16, marked between the red vertical lines, is the sample period used in this analysis.**

A notable feature of Figure 3 is the high cloud top $V_f$ for LWP0, which is greater by a factor of ~2 than any other

LWP bin. This discrepancy appears to be the result of a set of meteorological conditions in which the dissipation of an ice cloud generates fast falling ice crystals at cloud top. An example of such a dissipating ice cloud is depicted in Figure 4. By the 15[th] hour of the day, the cloud layer has properties that meet the requirements to be included in this study and IWC profiles are included (region indicated by red lines on Figure 4). During this period, cloud top height ranged from 1.410km to 1.815km, and cloud depth was between 0.27-0.99km. We suspect that large ice crystals are left behind at cloud top due to sublimation removing the smaller ice crystals. These large ice crystals have high values of $V_f$ at cloud top, and $V_f$ increases

marginally by cloud base because of a lack of available liquid water and vapor to contribute to ice mass growth. The cloud has limited, if any, liquid water so it is not possible for riming to add ice mass. Additionally, due to the high fall speeds, the residence time of the ice crystal in the cloud is likely to be low, thus minimizing depositional growth.




### 3.4 Environment influence on cloud IWC

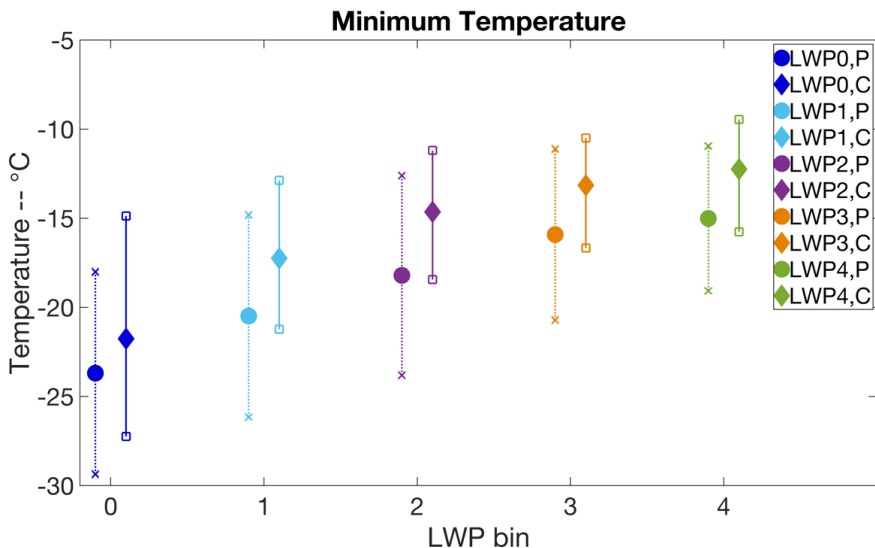

**Figure 5: Mean minimum cloud temperature for clean and polluted clouds for each LWP bin. Diamond markers with solid line indicates clean clouds, circle marker with dashed line represents polluted cases. The bars span the 20-80th percentile of the $T_{min}$**
5    **distribution for each bin.**

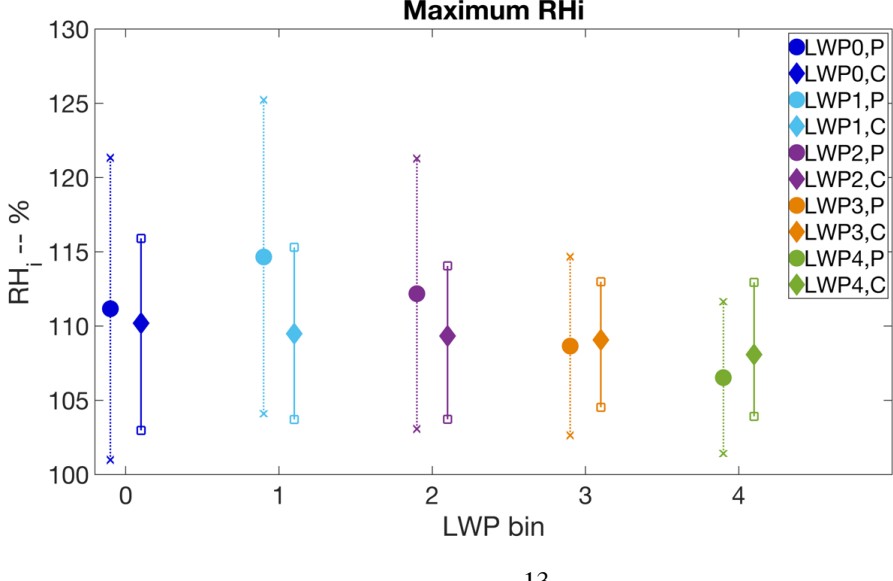





**Figure 6: Mean in cloud RHi for clean and polluted clouds for each LWP scenario. December is assigned the value 0. Diamond markers with solid line indicates clean clouds, circle marker with dashed line represents polluted cases. The bars span the 20-80th percentile of the $RHi_{max}$ distribution for each bin.**

The temperature and humidity properties of the environment in which a cloud forms influences the ice properties of the cloud. To define an aerosol alteration to the cloud microphysical state we first need to examine the impact the environment has on the cloud ice properties. Figures 5 and 6 present statistics of minimum cloud temperature ($T_{min}$), and maximum in-cloud relative humidity with respect to ice ($RHi_{max}$) of each LWP bin. While figure 7 depicts the distributions of sampled day of year for each LWP bin. Polluted clouds are consistently colder than clean clouds by $3 - 5°C$, which is likely due to

the seasonal dependence of the cloud sampling (Fig. 7). The colder temperatures found in the polluted cases leads to the expectation of increased ice crystal number because of the higher likelihood of ice nucleus activation at colder temperatures (DeMott et al., 2010). This is in addition to the possibility that higher INP concentrations are found in polluted environments, which would lead to the expectation of more nucleated ice crystals. Countering these effects is the fact that ice deposition rates are maximized at $-14.25°C$ near sea level when there is saturation with respect to liquid (Byers, 1965). For

the LWP0,1,2 bins, the minimum and mean temperatures in the cloud layer are closer to $-14.25°C$ in clean clouds than in polluted cases, see Table 1, which leads to the expectation of greater rates of depositional growth in the clean clouds in these cases. LWP3 has temperatures in clean and polluted clouds similarly favorable to depositional growth, while LWP4 temperatures suggest greater deposition rates in polluted clouds. With regard to RHi, the levels between clean and polluted clouds are comparable, with the low LWP bins, LWP0,1,2, having higher maximum RHi values in polluted cases, and the

high LWP bins, LWP3,4, have clean cases with slightly higher RHi values. Additionally, polluted cases have higher occurrences of extreme high RHi values – the ($RHi_{max}$) distributions have greater skewness towards elevated values.

| Minimum/Mean cloud layer Temperature ($°C$) | LWP0 | LWP1 | LWP2 | LWP3 | LWP4 |
|---|---|---|---|---|---|
| Clean | -21.8/-20.4 | -17.2/-15.7 | -14.7/-13.3 | -13.2/-12.0 | -12.3/-10.8 |
| Polluted | -23.7/-22.4 | -20.5/-18.8 | -18.2/-16.8 | -15.9/-14.8 | -15.0/-13.3 |

**Table 1: In-cloud Minimum temperature and mean cloud layer temperature for profiles in each LWP bin.**

Given the similarity between RHi levels and the minimum temperatures for clean and polluted clouds within each LWP bin, the distribution of ice crystal habits of the nucleated ice crystals in both cases should be similar. For LWP1,2,3,4 plate and dendrite type crystals are likely to be common, while LWP0 may be more apt to produce columns (Bailey and Hallett, 2009). For a given habit type, the reflectivity differences are dominated by variations in ice crystal size (Hong, 2007), and therefore the observed variations in measured radar reflectivity likely cannot be explained by habit effects alone.

More generally, the variation in temperature and supersaturation levels between clean and polluted clouds cannot explain the





observed differences in IWC. This further supports the notion that aerosols are altering the microphysical state of the cloud in manners which suppress ice mass production. These mechanisms are detailed in Section 4.

A few other features of the temperature and RHi distributions are interesting to note:

(1)   The mean temperature of the cloud layer increases with increasing LWP bin. This is likely due to the ability of warmer air masses to support higher levels of liquid water. Additionally, there may be a slight seasonal effect as the mean sample day (Fig. 7) of all the polluted and clean cases only vary slightly amongst the LWP bins. It is also interesting that there are few high LWP clouds found at relatively cold temperatures – LWP4 has few clouds with $T_{min} < -20°C$. The warm temperatures in these clouds also limit the level of RHi, suggesting that depositional ice mass growth may be limited in these cases despite the high levels of liquid water.

(2)   Likewise, there are more $RHi_{max} > 120\%$ in the low LWP bins. This suggest that supersaturation in these clouds is strongly temperature dependent, and less strongly controlled by the total amount of water contained within the cloud layer.

  (3)   While the RHi distributions for clean cases are fairly uniform across all the LWP bins, there is a high amount of variation in the RHi distributions for the polluted LWP bins. This could be the result of greater variability in the

meteorological conditions under which polluted clouds are found.

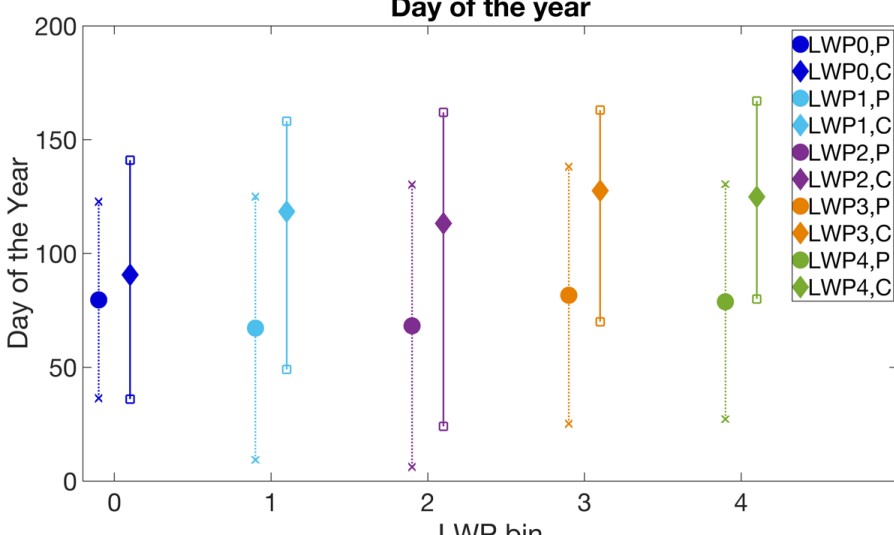

**Figure 7:** **Mean day of sample for the distribution of sampled profile for clean and polluted clouds for each LWP scenario. Diamond markers with solid line indicates clean clouds, circle marker with dashed line represents polluted cases. The bars span the 20-80[th] percentile of the day of year distribution for each bin.**




## 4 Discussion

### 4.1 Aerosol suppression of cloud ice nucleation

The observations presented in Sect. 3 indicate that polluted clouds have reduced amounts of cloud ice mass for a given amount of condensed liquid mass. At cloud top, nucleated ice crystals are commonly found in a highly saturated

environment with respect to ice, and therefore we expect ice nucleation and depositional processes to be the main controls on IWC in this region of the cloud. For polluted clouds the observed high $V_f$ is an indicator of larger mean ice crystals, relative to equivalent clean clouds. Given that the differences in IWC at cloud top are not statistically significant between clean and polluted clouds, the implication is that polluted clouds have reduced ice crystal number. That is, nucleation rate is suppressed.

In the Arctic mixed-phase clouds under study, heterogeneous freezing mechanisms are the primary source of nucleated ice crystals. Since these freezing processes are dependent on the liquid drop size distribution and the chemical composition of the solute laden liquid drops, it is plausible to assume higher aerosol concentrations within the cloud layer reduces the heterogeneous ice nucleation rate. Two mechanisms detailing how this may occur in these clouds are discussed here.

First, through the first indirect aerosol effect, the increase in aerosol concentration reduces both the mean droplet size and the width of the drop size distribution (Chandrakar et al., 2016). Hoffer (1961) offers evidence that larger liquid droplets freeze more readily than do small drops. The suppression of the ice nucleation rate through a reduction in the mean diameter of liquid droplets in mixed-phase clouds (e.g., Lance et al. 2011) could explain the observed reduction in ice crystal number at cloud top.

A second possibility is that polluted conditions could be leading to aerosol-aerosol interactions that reduce the effectiveness of INP. For example, inorganic aerosols can be coated by organics, and thus limit their effectiveness to nucleate ice (Diehl and Wurzler, 2004; Girard et al., 2005; Kulkarni et al., 2014). Variations in the winter and spring time scattering coefficient measurements used in this study are most strongly influenced by fluctuations in $SO_4^-$ sulfate aerosols (Quinn et al., 2002). Therefore, it is possible that the conditions we define as polluted are conditions in which INP are likely

to be coated by sulfates reducing the efficiency at which they nucleate ice. Through this lens, we would expect polluted conditions to be associated with a reduction in ice crystal nucleation rate.

        Our observations are consistent with simulations done by Girard et al. (2005), that show increasing sulfuric acid aerosols in Arctic clouds reduces ice crystal number concentrations while mean ice crystal size is increased. Other studies have found evidence for the ability of sulfates to suppress the onset of heterogeneous freezing (Eastwood et al., 2009), and

such inhibition results in the generation of fewer but larger ice crystals (Jouan et al., 2014). However, we currently do not have the measurements needed to determine which mechanism (CCN or INP) is the main control on ice nucleation in these clouds. We suspect that these two identified aerosol induced alterations to the ice nucleation rate would likely lead to different ice crystal size distributions. If INP alterations are the dominant mechanism, there is no reason to expect a size



dependence to the liquid drops that freeze. If CCN are the main modulator, a consistent size threshold for droplet freezing would be expected for both clean and polluted cloud since this second hypothesis only accounts for influences of droplet size on drop freezing, and not composition. This leaves the possibility for future observations to provide insight into the specifics of aerosol effects on ice nucleation. Observing in-cloud ice crystal size distributions, especially at small crystal sizes, would

provide insight into the size variability of nucleated ice crystals. This variability in ice crystal size should be linked to the in-cloud CCN or INP properties, with higher ice crystal size variability expected if INP are the dominate control on nucleation. We admit that this is a challenging measurement to make due to the fast depositional rate of ice when ice crystals are small, and so alternatively one could examine the effects on the liquid drop size distribution. The absence of large liquid drops would lend support to the idea that CCN are the main control of ice nucleation in Arctic mixed-phase clouds.

### 4.2 Secondary ice mass growth under varying aerosol conditions

The reduced nucleation rate in polluted clouds has implications for the total amount of depositional ice mass growth in the cloud layer. The ice mass deposition rate for an individual crystal is proportional to the inverse of the effective radius of that ice crystal for most crystal habits (Rogers and Yao, 1989). This implies that depositional growth will lead to convergence of

ice crystal sizes given sufficient time for growth to occur. In the clouds under study, we believe the in-cloud residence time of an ice crystal is greater than the time it takes for this size convergence to occur – see Appendix A. This suggests that IWC gained through deposition is strongly determined by initial crystal number, and not by initial crystal size. Thus, the higher ice crystal nucleation rates of clean clouds directly result in greater total amounts of depositional growth.

Deposition alone cannot explain all of the observed differences in IWC profiles. The highest LWP regime, LWP4,

is the only case in which ice crystals in clean clouds have greater $V_f$ than ice crystals in polluted clouds at cloud base. Here, we suspect liquid water and ice properties in clean clouds promote greater levels of riming, leading to the observed high fall speeds. Riming is an efficient mechanism for increasing fall speeds of larger ice crystals because unlike deposition, riming efficiency increases with ice crystal effective radius (Erfani and Mitchell, 2017). For a cloud layer, riming efficiency grows with ice crystal and liquid drop size, and it has been shown that riming efficiency is strongly related to the presence of large

liquid drops (Borys et al., 2003; Lohmann, 2004). Clean clouds are expected to have greater concentrations of efficiently collected large liquid droplets ($> 10\mu m$), along with higher numbers of ice crystals. Conversely, in polluted clouds production of both large liquid drops (Chandrakar et al., 2016) and ice crystals are suppressed. Therefore, for a volume of cloudy air, riming conditions are more favorable in clean clouds.

We expect the level of riming to be proportional to the amount of liquid water contained within the cloud. In the

LWP1,2 cases, the low amount of liquid water makes riming relatively less efficient and perhaps non-existent if the liquid drop distribution does not support riming. Therefore, we speculate that ice mass gains in these low LWP clouds are mainly occurring through depositional growth. These clouds also tend to have cold temperatures which promote the growth of dendritic crystal habits. Dendrite fall speeds are slow relative to other crystal types with similar mass (Kajikawa, 1974), and





therefore these ice crystals have long in-cloud residence times, enhancing depositional growth. Such depositional ice mass growth is consistent with the observed high cloud-base IWC and low $V_f$ of LWP1,2. For LWP3, we suspect that higher amounts of liquid water promote greater rates of riming. If there is sufficient liquid water, riming adds mass and alters the shape of the ice crystal such that it falls at greater velocities relative to its size (Jensen and Harrington, 2015). The higher $V_f$

reduces the ice crystal residence time, limiting depositional growth. The changes to ice crystal habit caused by riming can also reduce the rate of deposition (Jensen and Harrington, 2015). We speculate that in LWP3, which has an intermediate LWP level, ice mass growth through riming cannot compensate for the limited mass gain through deposition, and thus relatively lower cloud base IWC with a corresponding higher $V_f$ are observed (in relation to LWP1,2). In the LWP4 cases, high riming rates lead to fast-falling crystals and reduced cloud residence times. Mass gained through deposition is relatively

small, but this reduction in depositional growth is more than compensated for by the high levels of riming. This is consistent with observed high fall speeds at cloud base in the LWP4 case to go along with high IWC levels.

## 5 Conclusion

A 9-year record of ground-based observations of stratiform mixed-phase clouds from Utqiaġvik, Alaska was used in

conjunction with surface measurements of aerosol scattering coefficient to quantify the influence of aerosols on ice production in these clouds. Profiles of reflectivity, IWC and $V_f$ are normalized for cloud depth, and subsequently compared for clouds occurring under clean and polluted conditions. Generally speaking, clean clouds have greater reflectivity and IWC values throughout the majority of the cloud layer. It should be noted that there is a dependence of these variables on the LWP of the cloud, and this analysis attempts to control for the influence of liquid water on ice production. At cloud top,

where ice crystals tend to be small, the variation in IWC between clean and polluted cases is minimal. However, we suspect based on our observations that the clean aerosol state promotes more efficient ice mass growth processes (i.e. nucleation, deposition and riming) and therefore higher IWC in the lower regions of the cloud layer. We use the IWC information, in conjunction with $V_f$ profiles to gain insight into the physical mechanisms that lead to the observed disparity in IWC. We treat the problem of ice mass in two parts – nucleation of ice crystals, and growth through deposition and/or riming.

25        In regards to nucleation processes, our observations are consistent with two views of aerosol suppression of ice nucleation. First, in polluted clouds, aerosols reduce the occurrence of large liquid droplets, which in turn inhibits freezing because of the reduced tendency for small drops to freeze. Second, our measure of surface aerosols is strongly correlated with the presence of sulfates in the atmosphere. The higher solute levels found under polluted conditions may be interacting with potential INP, diminishing their nucleation efficiency. Determining which of these two mechanisms is responsible for

the suppression of ice nucleation would require knowledge of the size distributions of the nucleated ice crystals and liquid drops in addition to better measurements of in-cloud aerosol composition. Additionally, both mechanisms might be at play in




these clouds, further complicating the picture. Future research into this area will open the door for understanding exactly how aerosols are interacting within a cloud system to govern ice nucleation.

This paper then identifies how aerosols interact within the cloud system to affect the deposition and riming rates. For depositional processes, suppression of ice nucleation in polluted clouds reduces competition for available water vapor,
promoting rapid deposition of ice at cloud top. This idea is consistent with the observed high $V_f$ found in polluted clouds in the cloud top region. However, the total amount of ice mass growth through deposition is strongly dependent on the total surface area of ice present in the cloud, which is greater in clean clouds because of the higher ice crystal nucleation rate. Riming, on the other hand, is dependent on the number of ice crystals in addition to the liquid drop size distribution. Increasing CCN in polluted clouds reduces the effective radius of the liquid drops, which reduces riming efficiency, and in
turn decreases ice mass growth. The higher number of ice crystals and larger liquid drops prevalent in clean clouds result in an environment that is more favorable for riming processes to occur, particularly when LWP is high.

It is important to note that our analysis does not rely on direct knowledge of INP or CCN populations and we make no assumptions about how the scattering coefficient measurements represent INP/CCN levels. Instead, we have treated the problem of ice mass growth in mixed-phase clouds in relation to the general aerosol population as defined by the surface
measurements. In doing so, we have shown that ice mass growth is sensitive to the variations in the surface measured aerosol population. This study supports the hypothesis that the ice properties of a cloud are influenced by CCN and liquid phase processes. Having said this, to truly understand the relative roles of INP and alterations in the liquid properties of the cloud on ice nucleation and growth processes, a more advanced understanding of INP present in these mixed-phase cloud systems is needed.

Advances in this area will be required to truly constrain how a mixed-phase cloud interacts with the greater Arctic and global climate system. This coupling is largely tied to cloud phase composition, which is inherently linked to ice production mechanisms. The rate at which ice is produced in a mixed-phase cloud has direct consequences for the cloud macroscale properties, such as the cloud net radiative effect, lifetime and precipitation characteristics. Further work will enable a more detailed understanding of how aerosols alter in-cloud microphysics and the subsequent macrophysical
properties of these clouds -- necessary research for a complete view of the broader climate system.

**Appendix A – Depositional growth of ice crystals**

In the Sect. 3.6 we argued that deposition will cause nucleated ice crystals of varying size to converge to the same size in a time that is typically less than the residence time of an ice crystal in the cloud layer.

An approximation of the deposition rate is given by Rogers and Yao (1989):

$$\frac{dm}{dt} = \frac{4\pi C(S_i - 1)}{\left(\frac{L_s}{R_v T}\right)\frac{L_s}{KT} + \frac{R_v T}{e_i D_d}} \tag{A1}$$





where $m$ is the mass of the ice crystal, $C$ is a diffusion coefficient specific to the crystal habit, $S_i$ is the saturation ratio, $R_v$ is the individual gas constant of water vapor, $L_s$ is the latent heat of sublimation, $K$ is the coefficient of thermal conductivity of air, $T$ is the temperature of the air, $e_i$ is the vapor pressure over ice, and $D_d$ is the coefficient of diffusion of water vapor in air.

For a plate type ice crystal, $C = D/\pi$, where $D$ is the diameter of the ice crystal. Given a mass-diameter relation of $m = 2.0x10^{-2}D^3$, Eq. A1 can be integrated to yield:

$$D(t) = \sqrt{133.33t\frac{(S_i-1)}{\left(\frac{L_s}{R_vT}\right)\frac{L_s}{KT}+\frac{R_vT}{e_iD_d}} + D_0^2}$$

(A2)

where $D_0$ is the initial ice crystal diameter. We use Eq. A2 to calculate the time it takes for a nucleated ice crystal of size $D_0$ to grow to within 90% of the diameter of an ice crystal growing in the same environment with an initial diameter of $D_0 = 1.0mm$, for varying ice supersaturation levels. The results are shown in Table A1, with the coefficients in Eq. A2 as follows: $T = -10°C, D_d = 2.06x10^{-5}\,{}^{m^2}\!/_{s^{-1}}, K = 2.32x10^{-2}\,{}^{m^2}\!/_{s^{-1}}$.

| Crystal diameter, $D_0$, (mm) | Convergence time, $(S_i) = 2$ | Convergence time, $(S_i) = 10$ |
|---|---|---|
| $D_0 = 1$ | $\tau_{con90} = 0\ min$ | $\tau_{con90} = 0.0\ min$ |
| $D_0 = 0.5$ | $\tau_{con90} = 7.13\ min$ | $\tau_{con90} = 1.43\ min$ |
| $D_0 = 0.1$ | $\tau_{con90} = 10.18\ min$ | $\tau_{con90} = 2.05\ min$ |
| $D_0 = 0.01$ | $\tau_{con90} = 10.30\ min$ | $\tau_{con90} = 2.07\ min$ |
| $D_0 = 0.001$ | $\tau_{con90} = 10.30\ min$ | $\tau_{con90} = 2.07\ min$ |

**Table A1: The time for nucleated ice crystals to converge to within 90% of an ice crystal growing in the same environment but with an initial nucleated size of $1mm$, $\tau_{con90}$, for two supersaturation levels.**

      Equation A1 understates the depositional rate for small ice crystals at warmer temperatures ($T = 0° - 10°C$) (Fukuta, 1969), and therefore the convergence times found here are conservatively long. Regardless, depositional rate is

strongly dependent on the inverse of the diameter and so there is rapid convergence of crystal size, even for ice crystals with very small initial sizes.

      The other consideration is the ice crystal residence time in the cloud, $\tau_{res}$. A conservative estimate of $\tau_{res}$ is done by integrating over the mean $V_f$ profiles (i.e., black lines in Figure 3) for clouds of varying depths. The residence times are given in Table A2.



| Cloud Depth | $\tau_{res}$, Clean Clouds | $\tau_{res}$, Polluted Clouds |
|---|---|---|
| $300m$ | $15.5\ min$ | $14.2\ min$ |
| $500m$ | $25.8\ min$ | $23.6\ min$ |
| $1000m$ | $51.6\ min$ | $47.3\ min$ |

**Table A2: Computed cloud residence times for clean and polluted clouds for three cloud depths.**

Consistently, $\tau_{res} > \tau_{con90}$ for all combination of cloud depth and $D_0$ cases. This is evidence that nucleated ice crystals should converge through depositional growth to a near common size in clouds of most depths observed in this study. This estimate is conservative because it ignores cloud dynamic influences on ice crystal motion, such as updrafts, which would increase the residence time in cloud. We are confident that alterations to the cloud dynamics would not significantly impact the claim of expected ice crystal size convergence.

**Acknowledgements**. This research was conducted primarily under support from the US Department of Energy (DOE) Atmospheric System Research Program under grant DE-SC0013306. Additionally, support was provided by DOE grants DE-SC0011918, and DE-SC0008794 and the National Science Foundation under grant ARC 1203902. Cloud and atmosphere data products were obtained from the Atmospheric Radiation Measurement (ARM) Climate Research Facility, a US Department of Energy Office of Science User Facility sponsored by the Office of Biological and Environmental Research. Aerosol measurements were obtained from the National Oceanic and Atmospheric Administration's Earth System Research Laboratory – Global Monitoring Division. We also thank Jessie Creamean for useful discussions and support in the early stages of this of this project.

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
