# Peer review of "Observed aerosol suppression of cloud ice in low-level Arctic mixed-phase clouds"

_Atmospheric Chemistry and Physics, 2017_

## Referee Comment (RC1) · Anonymous Referee #2 · 26 Feb 2018

This study uses long term cloud radar and aerosol scattering observations to try and determine the effects of aerosols on the microphysical properties of mixed phase Arctic stratus. While this study is timely and sorely needed, I feel that their criteria for determining what is an ice cloud at the minimum are not well explained and need to be better justified as their criteria can easily include liquid clouds with drizzle or even larger cloud droplets. Since this affects most of their dataset I feel that this issue is the most important to address before I can accept this for publication. I also recommend the authors explain the aerosol indirect effects with more detail in the introduction and consider the possible role of secondary ice production processes such as the Hallett-Mossop process and, from more recent laboratory experiments, the formation of ice from spicules that form from frozen droplets.

[Figure]

Line 15: "ice nucleation" – do you mean reduced secondary production or reduced nucleation via increasing the liquid water content and therefore decreasing total available supersaturation?

Section 1, Paragraph 5, introduction: I think a more detailed introduction to the three indirect effects in mixed phase clouds are needed: the thermodynamic indirect effect, the glaciation indirect effect, and the riming indirect effect. Figure 1 of Jackson et al. (2012) provides a good summary of Lohmann and Feichter's three mixed phase indirect effects.

Section 1, Paragraph 6: This paragraph seems to be out of order and interrupts the flow of the paper. I think the information here belongs more to where you discuss how phase partitioning is critical, as it helps to justify why we need to study the phase partitioning of mixed phase clouds.

Section 2.1, Paragraph 1: What is the minimum detectable signal of this radar? A monodisperse size distribution of liquid drops with a concentration of 100 cm-3 and maximum dimensions of 20 microns (radius of 10 microns) should result in a reflectivity of about -20 dBZ, which is quite characteristic of the tops of single-layer arctic stratocumulus. Establishing approximately how small of particles the MMCR is sensitive to is critical as liquid cloud droplets have been observed in arctic stratus at temperatures as low as -30 degrees Celsius and I fear that observations of "small ice" that are pointed out in later sections could really be liquid drops.

Section 2.3. I think that the current criteria to eliminate as many liquid clouds as possible might be too simplistic. Liquid cloud particles can exist at temperatures as low 30 degrees Celsius, and even drizzle has been observed at temperatures as cold as -10 degrees Celsius. The authors need to better establish how sensitive the MMCR is to the smaller liquid particles, or perhaps only include regions that are subsaturated with respect to water but supersaturated with respect to ice in order to adequately ensure that they are only observing taking observations from ice in the clouds.

Page 8, line 14. These can easily be liquid droplets. Section 4.2. How much of an impact do you think the Hallett-Mossop process, active at temperatures from -3 to -8 degrees Celsius, would have in your higher LWP bin clouds in terms of secondary production? It may not necessarily be more riming, but there could also be more secondary ice crystals being produced by this process. Laboratory experiments have also shown that when droplets freeze they can produce spicules that then proceed to generate secondary ice crystals (see Lawson et al. 2015).

Figure 4. The color scale for reflectivity needs to be adjusted.

Figures 5, 6, 7. I found the figure legends difficult to understand with all of the entries and abbreviations. I would recommend revising the legends to make the figures easier to understand.

References: Rangno, A. L., and P. V. Hobbs (2001), Ice particles in stratiform clouds in the Arctic and possible mechanisms for the production of high ice concentrations, J. Geophys. Res., 106(D14), 15065–15075, doi:10.1029/2000JD900286.

Jackson, R. C., G. M. McFarquhar, A. V. Korolev, M. E. Earle, P. S. K. Liu, R. P. Lawson, S. Brooks, M. Wolde, A. Laskin, and M. Freer (2012), The dependence of ice microphysics on aerosol concentration in arctic mixed-phase stratus clouds during ISDAC and M-PACE, J. Geophys. Res., 117, D15207, doi:10.1029/2012JD017668.

Lawson, R.P., S. Woods, and H. Morrison, 2015: The Microphysics of Ice and Precipitation Development in Tropical Cumulus Clouds. J. Atmos. Sci., 72, 2429–2445, https://doi.org/10.1175/JAS-D-14-0274.1

---

## Referee Comment (RC2) · Anonymous Referee #1 · 21 Mar 2018

This is a review of the manuscript "Observed aerosol suppression of cloud ice in low-level Arctic mixed-phase clouds" by M. Norgren et al., submitted to ACPD. The authors analyze a 9-year long record of cloud and aerosol observations to describe in-cloud microphysical processes and the dependencies on the degree of pollution.

The results are interesting and highlight the further need to pin down the role of aerosols in determining the properties of cloud ice particles. The paper is well written in a mostly clear and organized way. In the following, I am raising a number of concerns which should be addressed prior to publication.

1) Throughout the manuscript, it is questionable to draw direct conclusions on the process of ice nucleation because nucleation alone may not be the only mechanism of initiating cloud ice. In particular when riming is assumed to be non-negligible, the po-

tential roles of rime splintering should be pointed out. In addition to that, mechanisms like droplet shattering upon freezing and upon ice-ice collisions have been under discussion recently. Overall, each of the ice-initiating mechanisms seems to be far from well-described or understood. Therefore, it feels more appropriate to refer to modified ice number concentrations only, rather than linking the indirect observations described here to exactly one of these processes. In fact, a recurring finding in studies of ice nucleation is that the concentrations of ice nuclei cannot explain ice number concentrations. Nevertheless, a discussion of potential mechanisms will be valuable at some point of the paper.

2) A point that remains unclear to me is related to the subsampling of data. The focus of this study are mixed-phase clouds, so the threshold for humidity is chosen to be 100% saturation with respect to ice. However, mixed-phase implies the presence of liquid droplets. In a mixed-phase cloud, humidity is usually close to water saturation (or higher in strong updrafts), otherwise the cloud droplets would evaporate quickly. Therefore it would be straightforward to choose water saturation, or a value close to that. Instead, the choice of RHi=100% explicitly includes humidities well below water saturation since temperatures are constrained to T<6°C. The authors indicate there may be also clouds with tiny or without any liquid in the LWP0 category, sometimes explicitly called ice clouds (e.g. page 11, line7). I suspect that such definitions may have a big impact on the results within this LWP bin, and indeed the properties seem to behave distinct in some ways. Therefore I strongly suggest to explore the impact of the threshold for humidity. Nevertheless, this kind of threshold may be problematic generally, since the saturation in clouds can be highly variable and is strongly tied to the structure of turbulent eddies, in particular with a small content of cloud droplets. So how meaningful is the profile of a single sounding which is supposed to represent intervals of 6 to 12 hours? Based on the manuscript I also cannot get an idea of how helpful the Mergesonde product is in addressing the problem of variability. Generally, to improve the clarity of the overall picture of mixed-phase, it might be beneficial to exclude ice-only clouds and introduce a lower threshold for LWP in the LWP0 category,

e.g., to exclude effects like sublimating small ice (see also below).

3)The discussion of results would benefit a lot by outlining the strategy of how the profiles of reflectivity, ice water content and fall speed will be interpreted. At this point it will be also helpful for the reader to explain what it means to use a reflectivity-weighted fall speed which actually represents the tail of the largest particles of the size distribution. For example (as on page 11, line 8), assume we compare two situations with the same IWC, but different Vf, where the latter difference would be caused only by the size distribution width. What is the measure of mean size and what would it mean in terms of the number difference? Otherwise, is the general assumption that the width of the size distribution is the same for both polluted and clean clouds and is it a good assumption?

Minor points

Page 2, line 25: suggest Barrow, Alaska

Page 3, line 12; Page 17 line 11: I recommend to rephrase "secondary ice mass growth" because it may be misleading and imply some connection to secondary ice production such as rime splintering. Personally I don't see a need to call growth secondary, assuming that any initiating process would not be "growth".

Page 5, line 1: Does the analysis ensure that only cloud decks are analyzed with a cloud fraction of 100% for a period considerably longer than the 120 min window? The statement on page 9, last line, seems to imply there would be lateral entrainment at the cloud boundaries.

Page 7, paragraph 1: To get a sense for the analyzed data, the total amount or fraction of analyzed days would be interesting.

Page 12, line 13: This is one of the points when I stumble over ice clouds rather than mixed-phase clouds while bothe the title and abstract make different statements. The effect of ice sublimation would be a clear indicator of lacking water, so why include such

situations?

Page 16, line 5: With droplet freezing being the primary nucleation mechanism, the saturation as such is not the relevant variable, but temperature is.

Page 16, line 7: How reliable are such conclusions about ice number when the estimate of IWC may have a relative error of 100%?

Page 16, line 16: Hoffer 1961 may be an appropriate reference for the freezing behavior of drizzle or rain drops in which the aerosol content would scale, more or less, with drop mass due to collisional growth. However, cloud droplets mainly grow by condensation, and thus will contain the same amount of aerosol during growth. This is different from Hoffer's method of producing particle-containing drops, while we expect that more aerosol surface area per drop would yield a higher chance of freezing.

Page 16, line 29: Due to the low temperatures investigated by Eastwood et al. 2009, it seems that this publication is hardly relevant for very most of the clouds summarized in Fig. 5. Also their humidities were mostly well below water saturation, while I am still assuming that the manuscript focusses on mixed-phase clouds.

Page 17, line 1: The statement on CCN is hard to understand, please rephrase.

Page 17, lin 14: typo: Yao

Figure 6: "December is assigned the value 0" might be showing up inadvertently, otherwise I do not understand.

---

## Author Comment (AC1) · 29 May 2018

The comment was uploaded in the form of a supplement:
https://www.atmos-chem-phys-discuss.net/acp-2017-1191/acp-2017-1191-AC1-supplement.pdf

---

## Author Comment (AC3) · 31 May 2018

We would like to thank both referees for their time and effort in reviewing this manuscript.

Review comments are in blue and the responses by the authors are in black text.

**Reviewer #1:**
1) Throughout the manuscript, it is questionable to draw direct conclusions on the process of ice nucleation because nucleation alone may not be the only mechanism of initiating cloud ice. In particular when riming is assumed to be non-negligible, the potential roles of rime splintering should be pointed out. In addition to that, mechanisms like droplet shattering upon freezing and upon ice-ice collisions have been under discussion recently. Overall, each of the ice-initiating mechanisms seems to be far from well-described or understood. Therefore, it feels more appropriate to refer to modified ice number concentrations only, rather than linking the indirect observations described here to exactly one of these processes. In fact, a recurring finding in studies of ice nucleation is that the concentrations of ice nuclei cannot explain ice number concentrations. Nevertheless, a discussion of potential mechanisms will be valuable at some point of the paper.

We agree with the point made by this comment that it is questionable to draw a clear and direct conclusion on ice nucleation processes based on the evidence we present in the paper. However, we feel these observations point to a shift in cloud microphysical processes due to the influence of aerosols. The manuscript, and specifically the Discussion section, has been reworked so that claims made about process level phenomena are more general in nature.

The Discussion section has been redone as follows:
- We put forth our inferences from the reflectivity and fall speed retrievals: among clean and polluted clouds, the cloud top reflectivity and IWC values are similar but the polluted clouds have significantly higher fall speeds. This is an indication that polluted clouds have fewer, but larger, hydrometeors. We then comment on why we feel ice is present in the clouds and why we feel we are not exclusively observing liquid only clouds. Our argument is that the mean Doppler velocities are relatively high which is suggestive of clouds containing ice particles. This is the main claim we are trying to show: that aerosols tend to coincide with mixed-phase clouds that have reduced ice crystal number concentrations and ice mass production. The discussion of the physics governing this suppression is of secondary importance because we do not have the appropriate measurements to pin down the details.

- We then list several physical mechanisms for producing cloud ice that could be altered by aerosols. We do not attempt to state which processes are responsible for the observed results. The list includes: depositional growth efficiency, riming efficiency, ice shattering and splintering (Hallet-Mossop properties).

- Yes, IN concentrations have been shown to be too low to fully explain the observed levels of ice in Arctic mixed-phase clouds, and therefore other ice crystal production methods must occur, However, these *secondary* methods (e.g. crystal shattering) are dependent on ice crystals being present and therefore, one would assume, on ice crystal number concentrations which are in part a function of the heterogeneous ice nucleation rate. It is therefore reasonable to assume that perturbations in the ice nucleation rate will be relevant to the overall ice crystal number found throughout the cloud layer. The complexities of ice production are an interesting question, and we hope that this paper, and specifically the Discussion section, contributes to the dialogue on how aerosols influence cloud ice properties.

Several reviewer comments related to claims made in the Discussion Section and we attempted to address these concerns through restructuring this section and focusing the paper more on the observed results of the higher IWC profiles found in clean clouds, and less on the physical mechanisms responsible for this shift in cloud IWC.

The Discussion section now reads as follows:

[revised manuscript text omitted]

2) A point that remains unclear to me is related to the subsampling of data. The focus of this study are mixed-phase clouds, so the threshold for humidity is chosen to be 100% saturation with respect to ice. However, mixed-phase implies the presence of liquid droplets. In a mixed-phase cloud, humidity is usually close to water saturation (or higher in strong updrafts), otherwise the cloud droplets would evaporate quickly. Therefore it would be straightforward to choose water saturation, or a value close to that. Instead, the choice of RHi=100% explicitly includes humidities well below water saturation since temperatures are constrained to T<6 ◦ C. The authors indicate there may be also clouds with tiny or without any liquid in the LWP0 category, sometimes explicitly called ice clouds (e.g. page 11, line 7). I suspect that such definitions may have a big impact on the results within this LWP bin, and indeed the properties seem to behave distinct in some ways. Therefore I strongly suggest to explore the impact of the threshold for humidity. Nevertheless, this kind of threshold may be problematic generally, since the saturation in clouds can be highly variable and is strongly tied to the structure of turbulent eddies, in particular with a small content of cloud droplets. So how meaningful is the profile of a single sounding which is supposed to represent intervals of 6 to 12 hours? Based on the manuscript I also cannot get an idea of how helpful the Mergesonde product is in addressing the problem of variability. Generally, to improve the clarity of the overall picture of mixed-phase, it might be beneficial to exclude ice-only clouds and introduce a lower threshold for LWP in the LWP0 category, e.g., to exclude effects like sublimating small ice (see also below).

We apologize for any confusion that the LWP0 (ice cloud) bin may have caused. The motivation for including clouds with little to no liquid water path, is to provide a null case that allows for greater contrast when examining the physical processes behind IWC production that become available when liquid water is present. That is, we are using LWP0 bin clouds as a basis for comparison between cloud types that have the potential for high rates of deposition and riming to those that do not.

If having LWP0 bin clouds remains an issue for the reviewer, we can easily remove these clouds from the analysis without significant alteration to the overall results of this paper. This LWP bin is a function of the uncertainty in the LWP retrieval from the microwave radiometer, which is roughly 15g/m2. We cannot say with a high probability

that any one cloud with a LWP below this threshold contains liquid water, and so at times, we refer to these as *ice clouds*. We have gone through the paper and removed instances where we call the LWP0 bin as ice clouds and instead refer to them as LWP0 bin clouds.

Regarding using relative humidity with respect to ice (RHi) as a designator throughout the paper, rather than with respect to liquid, we chose to use RHi because portions of this study deal with ice crystal nucleation and depositional growth processes. These physical processes are linked to the level of saturation with respect to ice in the cloud. Additionally, we have other (and better) methods of determining the presence of water in the cloud. Specifically, liquid water path is determined from microwave radiometer data. This is done for precisely the reason the reviewer mentions in the comment -- we do not have a sufficient data record (at the needed spatial or temporal resolution) on cloud humidity levels to determine saturation at the accuracy needed to make claims about the presence of liquid water. The cloud structure is too dynamic and variable to rely on 6-12 hour interpolations of radiosondes to determine humidity fields. The motivation for using RHi, is not to ensure a mixed-phase cloud, but rather it is used to gain a general view of the synoptic scale conditions that the cloud forms in. This paper does not rely on the mergesonde data to make strong claims, but rather we look for any differences in ice saturation across the complete sample of clouds to see if there is an obvious cross correlation that can explain observed differences in IWC. And finally, since ice supersaturation is the driver behind ice deposition, and therefore, we feel that it is the intuitive variable to use for defining the level of water vapor in the cloud layer.

We do have data for regions that are sub-saturated with respect to ice (for example, below cloud base), however we omitted these clouds from the study to simplify the set of physical processes that are occurring within the cloud layer. We only want to focus on deposition of ice, and less on clouds where sublimation is a significant feature.

3)The discussion of results would benefit a lot by outlining the strategy of how the profiles of reflectivity, ice water content and fall speed will be interpreted. At this point it will be also helpful for the reader to explain what it means to use a reflectivity-weighted fall speed which actually represents the tail of the largest particles of the size distribution. For example (as on page 11, line 8), assume we compare two situations with the same IWC, but different Vf, where the latter difference would be caused only by the size distribution width. What is the measure of mean size and what would it mean in terms of the number difference? Otherwise, is the general assumption that the width of the size distribution is the same for both polluted and clean clouds and is it a good assumption?

A hydrometeors ability to reflect radiation back to the radar is approximately proportional to the sixth power of its diameter. This non-linear relationship between particle size and reflectivity means that the largest particles in a sampled volume contribute the most signal to the reflectivity measurements. In the case of an ice crystal size distribution, given a fixed IWC, whether the distribution is broadening or shifting upward in size cannot be explicitly determined from the radar reflectivity measurements. However, an increase in reflectivity implies more large particles in both cases. For the results in this paper, the implication is that the increase in $V_f$, a reflectivity weighted fall speed, found in polluted clouds is the result of an increase in the presence of larger ice crystals. This, in conjunction with the IWC information, allows us to make broad claims about particle concentrations.

To provide the reader with more context for interpreting the results, the following paragraph has been added to the start of section 3:

*In this section we present mean in-cloud profiles of reflectivity, IWC and $V_f$ for the polluted and clean clouds found in each of the LWP bins. We use the relative relationships between IWC and $V_f$ amongst the clean and polluted clouds to make inferences about ice crystal size and number concentrations. This is followed by a discussion of possible microphysical processes within the cloud that may be causing the shifts in crystal size and concentration.*

To section 3.3, the following discussion has been added:

*Since the measured radar reflectivity scales approximately with the sixth power of hydrometer size, it is the largest hydrometers that will reflect the most radiation back to the radar. Thus, the reflectivity, and in turn the fall speed signal, are dominated by the largest hydrometers in the sampled volume. If a fixed amount of ice is sampled, it is not possible to determine if increases in reflectivity are due to an increase in the ice crystal size distribution, or a broadening of this distribution. However, the non-linear response of reflectivity to ice crystal size does mean that there is an increase in the presence of large ice crystals (sizes greater than the geometric mean). This knowledge about the relative populations of large ice crystals lets us make broad claims about ice crystal number concentrations in these clouds.*

Minor points:
Page 2, line 25: suggest Barrow, Alaska
Updated manuscript by adding the word *Alaska*, to reflect comment. Additionally, the Barrow name has been changed to Utqiaġvik, to reflect the current name of the city.

Page 3, line 12; Page 17 line 11: I recommend to rephrase "secondary ice mass growth" because it may be misleading and imply some connection to secondary ice

production such as rime splintering. Personally I don't see a need to call growth secondary, assuming that any initiating process would not be "growth".

Valid point and the passage has been reworded to omit the word "secondary".

Text now reads: *This includes aerosol influences on both nucleation of ice crystals and ice mass growth processes.*

Page 5, line 1: Does the analysis ensure that only cloud decks are analyzed with a cloud fraction of 100% for a period considerably longer than the 120 min window? The statement on page 9, last line, seems to imply there would be lateral entrainment at the cloud boundaries.

No, the analysis does not select for periods when cloud fraction is 100 percent. However, we did perform sensitivity analysis on the impacts of the averaging window time period on the ice crystal fall speed, $V_f$, and $V_f$ remained rather consistent. This consistency also indicates that the majority of the sampled cloud volumes did not come near cloud edge, and that the eddy structure contained within the clouds was variable. The statement at the end of Page 9 was referring to the possibility of vertical mixing. This sentence has been updated to reflect this:

*This decrease in the rate of IWC increase near cloud base is likely due to the impacts of less saturated air entraining vertically into the bottom of the cloud, slowing growth processes in this region.*

Page 7, paragraph 1: To get a sense for the analyzed data, the total amount or fraction of analyzed days would be interesting.

A table with summary statistics, including fractional occurrence of cloud type, has been added to this section of the paper.

Added table:

| LWP Bin | Mean Cloud depth, [m] | Mean LWP, [$gm^{-2}$] | Mean $T_{min}$, [°C] | Mean $RHi_{max}$, [%] | Number profiles |
|---------|----------------------|----------------------|----------------------|----------------------|-----------------|
| LWP0,C | 541 | 3.49 | −21.8 | 110.2 | 986 |
| LWP0,P | 580 | 3.61 | −23.7 | 111.1 | 1155 |
| LWP1,C | 380 | 17.85 | −17.2 | 109.5 | 832 |
| LWP1,P | 433 | 17.15 | −20.5 | 114.7 | 972 |
| LWP2,C | 372 | 43.22 | −14.7 | 109.3 | 3286 |
| LWP2,P | 395 | 39.87 | −18.2 | 112.2 | 2236 |
| LWP3,C | 448 | 109.07 | −13.2 | 109.1 | 2785 |
| LWP3,P | 422 | 91.51 | −15.9 | 108.6 | 1846 |
| LWP4,C | 548 | 213.31 | −12.3 | 108.1 | 1491 |

| LWP4,P | 492 | 244.35 | −15.0 | 106.5 | 654 |
| All Bins | 444 | 72.12 | −16.4 | 109.9 | 16244 |

**Table 1: Summary statistics for the sampled clouds in each LWP bin. The last row is the aggregate sample from all LWP bins.**

Page 12, line 13: This is one of the points when I stumble over ice clouds rather than mixed-phase clouds while both the title and abstract make different statements. The effect of ice sublimation would be a clear indicator of lacking water, so why include such situations?

We include ice clouds mainly as a set of clouds to reference the liquid containing clouds against. Ice clouds are used as a null case where riming, and other liquid dependent ice mass growth processes, are not a factor.

Page 16, line 5: With droplet freezing being the primary nucleation mechanism, the saturation as such is not the relevant variable, but temperature is.

This section of the paper has been reworked in response to other comments and the sentence is no longer in the paper. However, we were trying to say that, in regions of the cloud where riming is not a significant factor in IWC production, heterogeneous ice nucleation and deposition are the main controls of IWC. And while temperature is the main control on nucleation, the available water is a control on deposition.

Page 16, line 7: How reliable are such conclusions about ice number when the estimate of IWC may have a relative error of 100%?

We attempt to address the uncertainty of the IWC retrieval with the use of large samples of clouds that can be used to represent the cloud population. We preformed statistical test at a 95 percent confidence level to determine if differences between clean and polluted clouds exists.

Page 16, line 16: Hoffer 1961 may be an appropriate reference for the freezing behavior of drizzle or rain drops in which the aerosol content would scale, more or less, with drop mass due to collisional growth. However, cloud droplets mainly grow by condensation, and thus will contain the same amount of aerosol during growth. This is different from Hoffer's method of producing particle-containing drops, while we expect that more aerosol surface area per drop would yield a higher chance of freezing.

In the process of responding to major comment 1), the Hoffer reference has been removed. The discussion on the possible physical processes controlling ice growth in

clouds has been significantly altered to the point where we feel this comment has been sufficiently addressed.

Page 16, line 29: Due to the low temperatures investigated by Eastwood et al. 2009, it seems that this publication is hardly relevant for very most of the clouds summarized in Fig. 5. Also their humidities were mostly well below water saturation, while I am still assuming that the manuscript focuses on mixed-phase clouds.

This a good point, and the fact that the Eastwood et al. paper deals with conditions that are not saturated with respect to liquid, was an oversight on our part. However, there is a case in the Eastwood et al. paper in which sulfates require that there be water saturation for ice nucleation to occur. This does not provide absolute evidence for our claim, but it does support it.

Page 17, line 1: The statement on CCN is hard to understand, please rephrase.

This section of the text has been rewritten and in that process this statement was removed.

Page 17, line 14: typo: Yao
Corrected

Figure 6: "December is assigned the value 0" might be showing up inadvertently, otherwise I do not understand.

This is a typo and has been removed.

**Reviewer #2:**

While this study is timely and sorely needed, I feel that their criteria for determining what is an ice cloud at the minimum are not well explained and need to be better justified as their criteria can easily include liquid clouds with drizzle or even larger cloud droplets. Since this affects most of their dataset I feel that this issue is the most important to address before I can accept this for publication.

This general comment is addressed in the response to the following comments.

Line 15: "ice nucleation" – do you mean reduced secondary production or reduced nucleation via increasing the liquid water content and therefore decreasing total available Supersaturation?

In response to another comment, the term *secondary production* has been removed from Line 15. In this paper, ice nucleation only refers to the generation of a new ice crystal through homogeneous or heterogeneous nucleation, and it does not include rime splintering or any other secondary ice production processes.

The sentence containing line 15 has been altered to the following:
*We additionally analyze radar-derived mean Doppler velocities to better understand the drivers behind this relationship, and conclude that aerosol induced reduction of the ice crystal nucleation rate, together with decreased riming rates in polluted clouds, are likely influences on the observed reductions in IWC.*

Section 1, Paragraph 5, introduction: I think a more detailed introduction to the three indirect effects in mixed phase clouds are needed: the thermodynamic indirect effect, the glaciation indirect effect, and the riming indirect effect. Figure 1 of Jackson et al. (2012) provides a good summary of Lohmann and Feichter's three mixed phase indirect effects.

We agree that providing a more thorough background in aerosol indirect effects strengthens the paper and we have added the following to paragraph 5 of section 1:

*That being said, several aerosol-cloud effects have been detected in mixed-phase cloud systems: the first and second aerosol indirect effects have been observed (Lohmann and Feichter, 2005). These two aerosol indirect effects, associated with the liquid phase of cloud, lead to further aerosol-induced implications in mixed-phase clouds. The thermodynamic indirect effect, whereby the reduced mean liquid drop diameter caused by increasing CCN makes cloud conditions less favorable to secondary ice production (e.g. rime splintering, contact nucleation), has the effect of reducing IWC in mixed-phase clouds with high CCN levels. Similarly, the riming indirect effect, the process in which CCN reduce the liquid drop size distribution so that the liquid drops are less efficiently collected by falling ice crystals, reducing the riming rates within a mixed-phase cloud (Borys et al., 2003). The reduced riming rate decreases ice production, and lowers cloud IWCs. Finally, the glaciation indirect effect, in which an increase in aerosols (traditionally INP from black carbon) is associated with greater levels of ice nuclei, which promotes greater conversion of liquid to ice within the mixed-phase cloud layer (Lohmann, 2002). Yet the specifics of how these cloud processes play out over time to determine the macroscale properties of clouds is poorly understood.*

*Several observational studies have found evidence for aerosol impacts on Arctic mixed-phase clouds. Using surface-based sensors at Barrow, both Garrett and Zhao (2006)*

*and Lubin and Vogelmann (2006) showed that a reduction of droplet size associated with elevated aerosol particle concentrations results in elevated emissivity of the cloud layer, thereby significantly increasing longwave radiation at the surface and contributing to warming. Lance et al. (2011) used in situ data from Arctic clouds to show that CCN concentrations, through the first indirect effect and riming indirect effect, may have a stronger influence on ice production than do INP concentrations. These past studies suggest that further interrogation of aerosol alterations to the microphysical state of mixed-phase clouds systems is warranted.*

Section 1, Paragraph 6: This paragraph seems to be out of order and interrupts the flow of the paper. I think the information here belongs more to where you discuss how phase partitioning is critical, as it helps to justify why we need to study the phase partitioning of mixed phase clouds.

Agreed, and the manuscript has been changed so that what was Paragraph 6 is now Paragraph 3.

Section 2.1, Paragraph 1: What is the minimum detectable signal of this radar? A monodisperse size distribution of liquid drops with a concentration of 100 cm-3 and maximum dimensions of 20 microns (radius of 10 microns) should result in a reflectivity of about -20 dBZ, which is quite characteristic of the tops of single-layer arctic stratocumulus. Establishing approximately how small of particles the MMCR is sensitive to is critical as liquid cloud droplets have been observed in arctic stratus at temperatures as low as -30 degrees Celsius and I fear that observations of "small ice" that are pointed out in later sections could really be liquid drops.

We agree that it was irresponsible to make strong claims about the phase of ice particles (e.g. "small ice") at cloud top and we have altered the manuscript to reflect the uncertainty in determining cloud phase in this region of the cloud layer.

We have also made significant changes to the Discussion section that address many of the issues raised by this comment.

The MMCR has a sensitivity down to roughly -50 dBZ, and we fully expect the radar to be observing liquid (in addition to ice) in the cloud top region. While we do not have the ability to directly determine the phase of the hydrometers from the radar, we feel the IWC and fall speed profiles strongly suggest the presence of ice formation at cloud top. Both the IWC and $V_f$ profiles have a significant and continuous increases through the cloud layer to cloud base, which is consistent the generation of ice crystals at cloud top, that fall through the liquid layer and undergo ice mass growth processes. Additionally, at

cloud top, the mean Doppler velocity (MDV) does provide some insight into phase, where the observed MDV values are $\sim 0.2 ms^{-1}$, and these high downward fall speeds are evidence for ice. If the radar reflectivity signal is dominated by liquid droplets, we expect MDV to be very close to $\sim 0 ms^{-1}$.

Furthermore, if we are to assume that we are observing clouds that contain only liquid and no ice, the observed aerosol effects on the liquid drop distributions are counter to what is predicted by the first aerosol indirect effect. Traditionally, with increasing aerosol levels, one expects greater number concentrations of liquid drops with reduced effective radius. Yet, in the clouds observed in this study, the cloud top values of hydrometeor fall speed tend to be greater in the polluted cases, which implies the presence of larger hydrometeors. The microphysics of these clouds do not resemble the typical physics of liquid clouds. We feel that this is basic evidence for a more complex system that contains ice.

Drizzle adds more complexity to phase classification from the radar data because drizzle would have MDV values similar to falling ice crystals. Though, at the minimum cloud temperatures used in this study (T<-6 deg. C) we feel that there is a limited impact from drizzle events. Drizzle events will be restricted to the warmer clouds in our data set because the mean minimum cloud temperatures for all LWP bins are well below -10 degrees Celsius (see Figure 5). Moreover, studies have shown that Arctic clouds found in the lower 2km of the atmosphere during the months of December through May, frequently contain both liquid water and ice (Shupe et al., 2005). And mixed-phase clouds are more prevalent at Utqiaġvik than are liquid clouds during the months under consideration in this study (Liu et al., 2017).

Section 2.3. I think that the current criteria to eliminate as many liquid clouds as possible might be too simplistic. Liquid cloud particles can exist at temperatures as low 30 degrees Celsius, and even drizzle has been observed at temperatures as cold as -10 degrees Celsius. The authors need to better establish how sensitive the MMCR is to the smaller liquid particles, or perhaps only include regions that are subsaturated with respect to water but supersaturated with respect to ice in order to adequately ensure that they are only observing taking observations from ice in the clouds.

This comment is closely related to the previous comment where we outline our case for why we feel the radar reflectivity signal is dominated by the ice phase, and not the liquid phase of the cloud hydrometeors.

We are concerned with regions of the cloud that are saturated with respect to water and ice because we are interested in how the liquid properties of the cloud influence cloud

ice production. Selecting clouds that are subsaturated with respect to water will limit the occurrences of mixed-phase clouds, which are the main subject of this study.

To avoid confusion about our motives in studying clouds containing both liquid and ice, the following sentence has been added to paragraph 1 of Sect. 2.3:
*We are interested in the interaction between the liquid and ice phase hydrometeors in the cloud and therefore we investigate ice in mixed-phase clouds that are saturated with respect to both liquid and ice.*

Page 8, line 14. These can easily be liquid droplets. Section 4.2. How much of an impact do you think the Hallett-Mossop process, active at temperatures from -3 to -8 degrees Celsius, would have in your higher LWP bin clouds in terms of secondary production? It may not necessarily be more riming, but there could also be more secondary ice crystals being produced by this process. Laboratory experiments have also shown that when droplets freeze they can produce spicules that then proceed to generate secondary ice crystals (see Lawson et al. 2015).

We agree that we should not be as specific in stating what physical processes are controlling ice production in clouds. In response to this and other comments, Section 4.2 has been altered to include a broader view of secondary ice production and how it may relate to the clouds observed in this study.

Regarding the Hallett-Mossop process, we expect this could play a role in ice crystal production in the warmer clouds included in this study. Though unlike the Lawson et al. paper mentioned by the reviewer, we are studying Arctic clouds which tend to be in a cleaner environment than the tropics. We expect the dearth of IN found in the Arctic to make Hallett-Mossop, and other secondary ice production mechanisms, a possible control of the ice mass budget in these clouds. This has now been addressed in the updated Discussion section:
*Moreover, when riming occurs there is rime splintering, a process that generates small ice crystals when a liquid drop is collected by an existing ice crystal (Hallett and Mossop, 1974). Rime splintering increases the ice crystal number concentration and it occurs more commonly at warmer temperatures (-3 to-8℃). This may help to explain why LWP4 bin clouds have the highest observed levels of IWC, despite these clouds tendency to be warm.*

Figure 4. The color scale for reflectivity needs to be adjusted.

Color scale has been altered and there is an updated figure in paper.

Figures 5, 6, 7. I found the figure legends difficult to understand with all of the entries and abbreviations. I would recommend revising the legends to make the figures easier to understand.
Legends have been reformatted.

---

## Editor Decision (ED1)

Here are a list of comments from the editor:

1. Page 3, Line 1. "thermodynamic effect". Please give one or two references for this effect. Line 3. Does "contact nucleation" really belong to secondary ice production? In my opinion, this mechanism belongs to primary ice production.
2. Page 3, Line 19. "aerosol alterations of  cloud liquid properties". I don't think that this statement is precise. Aerosol induced reduction of ice nucleation rate, as one of the main reasons for the changes of IWC claimed in this study, is not actually related to the alterations of cloud liquid properties.
3. Page 6, Line 16. "...in mixed-phase clouds that are saturated with respect to both liquid and ice". It is unclear whether you are investigating mixed-phase clouds saturated with respect to liquid or ice? In the next sentence, you are using the relative humidity with respect to ice to clarify the mixed-phase clouds.
4. Page 11, Line 19. "an increase in the ice crystal size distribution". What do you mean for this? Increase in geometric mean size?
5. Page 14, Line 9. "...which is likely due to the seasonal dependence of the cloud sampling (Fig. 7)". There is no information shown in Figure 7 of the seasonal dependence of the sampling. It is interesting to note that the polluted clouds tend to be a few degrees colder than clean clouds. This will result in more ice production and higher ice number. What is the implication of this for the IWC differences between polluted and clean clouds? Please discuss.
6. Page 15, Line 6. "Given the similarity between RHi levels and the minimum temperatures for clean and polluted clouds within each LWP bin," However, the minimum temperatures are actually quite different between clean and polluted clouds within each LWP bin.
7. Page 16, Line 8. "between cloud types". This is first time in the manuscript that "cloud types" is mentioned and it may cause the confusion. Do you mean that polluted and clean clouds belong to different cloud types?
8. Page 17, Line 10. "$SO_4^-$ sulfate". Remove "$SO_4^-$ ".
9. Page 20, Line 15. "cloud phase composition". What do you mean cloud phase composition? Phase partitioning or cloud phases (then remove "composition")
10. Page 21, Line 7 in Table A1. What is the unit of Si=2, 10? Shall add "%"?

---

## Author Response (AR2)

We thank the editor for their time and effort in reviewing this manuscript. The editor's comments are in blue colored text, responses are in black text, and alterations to the manuscript are in italicized text with red indicating areas that have been changed.

1. Page 3, Line 1. "thermodynamic effect", Please give one or two references for this effect.

   A reference to Lohmann and Feichter (2005), has been added to the part of this paragraph that deals with the thermodynamic indirect effect.

   *The thermodynamic indirect effect, whereby the reduced mean liquid drop diameter caused by increasing CCN makes cloud conditions less favorable to secondary ice production (e.g. rime splintering, contact nucleation), has the effect of reducing IWC in mixed-phase clouds with high CCN levels (Lohmann and Feichter, 2005).*

   Line 3. Does "contact nucleation" really belong to secondary ice production? In my opinion, this mechanism belongs to primary ice production.

   Secondary ice production "is a mechanism or process that produces new ice crystals in the presence of preexisting ice without requiring the action of an ice nucleating particle (or homogeneous freezing)." -- Field et al., 2017

   We agree with the editor that contact nucleation is not typically defined as a secondary ice production mechanism. In the text. "contact nucleation" has been exchanged for "collision fragmentation".

   *The thermodynamic indirect effect, whereby the reduced mean liquid drop diameter caused by increasing CCN makes cloud conditions less favorable to secondary ice production (e.g. rime splintering, collision fragmentation), has the effect of reducing IWC in mixed-phase clouds with high CCN levels.*

2. Page 3, Line 19. "Aerosol alterations of cloud liquid properties". I don't think that this statement is precise. Aerosol induced reduction of ice nucleation rate, as on of the man reasons for the changes of IWC claimed in this study, is not actually related to the alterations of cloud liquid properties.

   We agree with the editor that this statement does not properly align with the conclusions made about IWC production later in this paper. The point being outlined in the original text is that the IWC in the observed mixed-phase clouds have a response to the amount of surface aerosols present. Whether the changes in IWC are due to alterations to the CCN or the INP populations is unclear. The original sentence errored in being too specific. The first two sentences of the paragraph have been revised as follows:

   *In this paper, we aim to demonstrate that aerosol interact with Arctic mixed-phase cloud systems in ways that control ice crystal nucleation rates, as well as ice mass growth*

*processes.*

3. Page 6, Line 16. "...in mixed-phase clouds that are saturated with respect to both liquid and ice". It is unclear whether you are investigating mixed-phase clouds saturated with respect to liquid or ice? In the next sentence, you are using the relative humidity with respect to ice to clarify the mixed-phase clouds.

This study is investigating mixed-phase clouds that are saturated with respect to ice, and we do not explicitly require saturation with respect to liquid. For a sampled vertical cloud profile to be included in this study, it must be saturation with respect to ice at some location within its profile. This is one of several methods (along with minimum cloud profile temperature) that we use to ensure that the sampled clouds are mixed-phase, rather than being purely liquid clouds. To constrain the liquid properties of the clouds, we use liquid water path retrievals from a microwave radiometer (see Sect. 2.1 for details). The end of the paragraph containing this sentence was confusing and poorly ordered. We have removed the sentence and rewrite lines 14-20 of page 6 as follows:

*The corresponding temperature and humidity profiles for each IWC profile are identified from the mergesonde data, and they are used to characterize the maximum in-cloud relative humidity with respect to ice ($RH_i$) and minimum temperature ($T_{min}$) of each individual profile. The $RH_i$ information is used to infer which ice mass growth processes are available within the cloud layer (i.e. if deposition is possible). The minimum temperature data is used to improve the IWC retrieval. Since the IWC retrieval does not explicitly select for the presence of ice, it risks contamination from liquid water at warmer temperatures. To limit the occurrence of this contamination, we select for cloud profiles with $T_{min} < -6°C$, though it is possible that some of these clouds may still be lacking ice.*

4. Page 11, Line 19. "an increase in the ice crystal size distribution". What do you mean for this? Increase in geometric mean size?

Yes, we are referring to an increase in the geometric mean size of the ice crystals. As the crystals gain mass, their physical dimensions grow which we define as an increase in geometric mean size. The geometric mean size is related to the terminal velocity of the ice crystal, and so we can use ice crystal fall speeds to infer information about this size property. The sentence has been clarified:

*Considering the equivalent IWCs at cloud top, the greater fall speeds in the polluted clouds indicates the presence of ice crystals with larger geometric mean size, which must be matched with a reduction in ice crystal number, to drive the observed reflectivity/IWC response (a more detailed discussion is offered in Section 4.1/4.2).Alternatively, the fall speed variation could be the result of aerosol-induced changes in crystal habit and orientation, though we do not have evidence that these properties are influenced by INP concentrations.*

*Since the measured radar reflectivity scales approximately with the sixth power of hydrometer size, it is the largest hydrometers that will reflect the most radiation back to the radar. Thus, the reflectivity, and in turn the fall speed signal, are dominated by the largest hydrometers in the sampled volume. If a fixed amount of ice is sampled, it is not possible to determine if increases in reflectivity are due to an increase in the mean of the geometric size distribution of the ice crystals, or a broadening of this distribution.*

5. Page 14, Line 9. "...which is likely due to the seasonal dependence of the cloud sampling (Fig. 7)". There is no information shown in Figure 7 of the seasonal dependence of the sampling. It is interesting to note that the polluted clouds tend to be a few degrees colder than clean clouds. This will result in more ice production and higher ice number. What is the implication of this for the IWC differences between polluted and clean clouds? Please Discuss.

In reference to the seasonal dependence comment -- in Figure 7, for each LWP bin, the day of year distributions for clean clouds are all shifted to later in the year in comparison with to the sampled polluted clouds. The sentence in the manuscript has been altered to provide better direction for where to look for the seasonal dependence.

*Polluted clouds are consistently colder than clean clouds by 3 − 5°C, which is likely due to the seasonal dependence of the cloud sampling (see Fig. 7).*

In regards to the temperature difference between clean and polluted clouds -- the polluted clouds tend to be a few degrees colder, on average, than the clean clouds. This leads us to expect greater availability of ice nucleating particles, as well as higher rates of ice deposition onto existing ice crystal surfaces. These two factors imply that polluted clouds should have higher levels of IWC, *yet* our observations indicate the opposite. We feel this is strong evidence that systematic differences in the environments that clean and polluted clouds are found in are not the cause of the observed IWC differences. Rather aerosols are causing microphysical changes to the clouds systems that work in a manner to inhibit ice production in polluted clouds. In the Discussion section of the paper, we outline two physical mechanisms that may be the reason behind the low levels of IWC in polluted clouds. They are (1) reductions in efficiency of the ice nuclei due to coating of INP with hydrophilic compounds (such as sulfates), and (2) the lower ice crystal number that is the result of the reduced nucleation rate leads to less deposition and riming of ice. All of which reduce the IWC levels of polluted clouds.

6. Page 15, Line6. "Given the similarity between RHi levels and the minimum temperatures for clean and polluted clouds within each LWP bin," However, the minimum temperatures are actually quite different between clean and polluted clouds within each LWP bin.

We agree that this was poor phrasing because while the RHi levels are similar, the minimum temperatures are a few degrees lower for polluted clouds. In the context of the

paragraph, which is discussing the impact of these differences on ice crystal habit, the temperature variations are not large enough to make a difference in the ice crystal habit (Bailey and Hallet, 2009). The sentence has been redone for clarity:

*Given the similarity* *in* *RHi levels*, *and* *that* *the differences in minimum temperatures for clean and polluted clouds within each LWP bin* *are minor enough to not significantly alter ice crystal habit properties (Bailey and Hallet, 2009),* *the distribution of ice crystal habits of the nucleated ice crystals in both cases should be similar.*

7. Page 16, Line 8. "Between cloud types". This is the first time in the manuscript that "cloud types" is mentioned and it may cause the confusion. Do you mean that polluted and clean clouds belong to different cloud types?

The term "cloud types" is only referring to clean and polluted clouds, and does not carry any other meaning in this context. This instance is the terms only use in the manuscript and we agree that it easily leads to confusion. "Cloud types" has been removed from the text and clean and polluted clouds have been written out explicitly.

*In the cloud top region of polluted clouds, the high* $v_f$ *signifies larger hydrometeors, and since the differences in IWC and reflectivity here are not statistically significant between*  *clean and polluted clouds, the implication is that polluted clouds have a reduction in hydrometeor concentration.*

8. Page 17, Line 10. "$SO_4^-$ sulfate". Remove "$SO_4^-$".

Done.

*Variations in the winter and spring time scattering coefficient measurements used in this study are most strongly influenced by fluctuations in*  *sulfate aerosols (Quinn et al., 2002).*

9. Page 20, Line 15. "cloud phase composition". What do you mean cloud phase composition? Phase partitioning or cloud phases (then remove "composition").

Yes, we simply mean phase partitioning of mixed-phase clouds is a variable that influences how these clouds interact in the broader Arctic and global climate systems. The word "composition" has been removed from the sentence.

*This coupling is largely tied to cloud phase* *, which is inherently linked to ice production mechanisms.*

10. Page 21, Line 7. In Table A1. What is the unit of SI=2? Shall add "%"?

These supersaturations are in percent. $S_i = 1.02$is a 2% supersaturation. Table A1 has been updated to include "%" signs.

[revised manuscript text omitted]